# Shank is a dose-dependent regulator of Ca$_v$1 calcium current and CREB target expression

Edward Pym[1,2], Nikhil Sasidharan[1,2], Katherine L Thompson-Peer[1,2†], David J Simon[1,2,3‡], Anthony Anselmo[1], Ruslan Sadreyev[1], Qi Hall[1,2], Stephen Nurrish[1,2], Joshua M Kaplan[1,2,3*]

[1]Department of Molecular Biology, Massachusetts General Hospital, Boston, United States; [2]Department of Neurobiology, Harvard Medical School, Boston, United States; [3]Program in Neuroscience, Harvard Medical School, Boston, United States

**Abstract** Shank is a post-synaptic scaffolding protein that has many binding partners. Shank mutations and copy number variations (CNVs) are linked to several psychiatric disorders, and to synaptic and behavioral defects in mice. It is not known which Shank binding partners are responsible for these defects. Here we show that the *C. elegans* SHN-1/Shank binds L-type calcium channels and that increased and decreased *shn-1* gene dosage alter L-channel current and activity-induced expression of a CRH-1/CREB transcriptional target (*gem-4* Copine), which parallels the effects of human Shank copy number variations (CNVs) on Autism spectrum disorders and schizophrenia. These results suggest that an important function of Shank proteins is to regulate L-channel current and activity induced gene expression.

*For correspondence: kaplan@molbio.mgh.harvard.edu

Present address: † Department of Physiology, University of California, San Francisco, San Francisco, United States; ‡Department of Neurobiology, Stanford School of Medicine, Palo Alto, United States

Competing interests: The authors declare that no competing interests exist.

## Introduction

Shank is a synaptic scaffolding protein (containing SH3, Ankyrin, PDZ, proline-rich and SAM domains) (*Grabrucker et al., 2011*). Because Shank is highly enriched in the post-synaptic densities of excitatory synapses, prior studies have focused on the idea that Shank proteins regulate some aspect of synapse formation or function. Through its various domains, Shank proteins bind hundreds of other synaptic proteins (*Lee et al., 2011*; *Sakai et al., 2011*) thereby potentially altering diverse cellular functions. Shank proteins have been implicated in synaptic transmission, synapse formation, synaptic plasticity, and cytoskeletal remodeling (*Jiang and Ehlers, 2013*).

Mammals have three Shank genes, each encoding multiple isoforms (*Jiang and Ehlers, 2013*). Several mouse Shank knockouts have been described but these mutants exhibit inconsistent (often contradictory) synaptic and behavioral defects (*Jiang and Ehlers, 2013*), most likely resulting from differences in which Shank isoforms are impacted by each mutation. The biochemical mechanism by which Shank mutations alter synaptic function and behavior has not been determined.

In humans, Shank mutations and CNVs are linked to Autism Spectrum Disorders (ASD), schizophrenia, and mania (*Durand et al., 2007*; *Peça et al., 2011*). Haploinsufficiency for 22q13 (which spans the Shank3 locus) occurs in Phelan-McDermid syndrome (PMS), a syndromic form of ASD (*Phelan and McDermid, 2012*). PMS patients exhibit autistic behaviors accompanied by hypotonia, delayed speech, and intellectual disability (ID) (*Bonaglia et al., 2011*). Heterozygous inactivating Shank3 mutations are found in sporadic ASD and schizophrenia (*Durand et al., 2007*; *Peça et al., 2011*). These genetic studies suggest that decreased Shank3 function likely plays an important role in the pathophysiology of these psychiatric disorders.

A parallel set of genetic studies suggest that increased Shank3 function also contributes to psychiatric diseases. 22q13 duplications spanning Shank3 are found in ASD, schizophrenia, ADHD, and bipolar disorder (*Durand et al., 2007*; *Failla et al., 2007*; *Han et al., 2013*). These 22q13 duplications involve multiple genes; consequently, the contribution of increased Shank3 to these psychiatric disorders was uncertain. To address this issue, a transgenic mouse model was developed that selectively over-expresses Shank3 (*Han et al., 2013*). This transgenic mouse exhibited hyperactive behavior and susceptibility to seizures. Taken together, these studies suggest that too little or too much Shank3 is associated with several psychiatric disorders.

If Shank3 mutations and CNVs are causally associated with these psychiatric disorders, cellular and circuit phenotypes should also be sensitive to Shank3 copy number. Consistent with this idea, several defects have been reported in Shank3$^{+/-}$ heterozygotes, including: decreased mEPSC frequency and spine density (*Zhou et al., 2016*), decreased $I_h$ current density (*Yi et al., 2016*), decreased TRPV1 current density (*Han et al., 2016*), increased tactile sensitivity (*Orefice et al., 2016*), and decreased post-synaptic Homer abundance (*Wang et al., 2016*). Increased Shank expression was associated with increased spine density in hippocampus and decreased inhibitory synapses (*Han et al., 2013*). In many cases (*Han et al., 2013*; *Orefice et al., 2016*; *Wang et al., 2016*; *Zhou et al., 2016*), it was not determined if these phenotypes are a cell autonomous consequence of altered Shank3 copy number. While these studies identify cellular deficits associated with Shank3 CNVs, it remains unclear which Shank binding partners and cellular functions are responsible for psychiatric traits, nor why these traits are sensitive to both increased and decreased Shank gene dosage.

To further investigate how Shank proteins regulate nervous system development and function, we analyzed Shank function in an invertebrate genetic model. Here we show that *C. elegans* SHN-1 is a dose-sensitive regulator of Ca$_v$1 calcium current and CREB induced gene expression in *C. elegans* body muscles.

## Results

### The SHN-1 PDZ domain binds EGL-19/Ca$_v$1 channels

*C. elegans* has a single Shank gene, *shn-1*. The SHN-1 protein lacks an SH3 domain but has all other domains found in mammalian Shank proteins (*Figure 1A*). Many protein ligands have been identified for the Shank PDZ domain (*Lee et al., 2011*). Of these potential binding partners, we focused on EGL-19/Ca$_v$1 because human CACNA1C (which encodes a Ca$_v$1 α-subunit) is mutated in Timothy Syndrome (TS), a rare monogenic form of ASD (*Splawski et al., 2005*, *2004*), and polymorphisms linked to CACNA1C are associated with multiple psychiatric disorders (*Cross-Disorder Group of the Psychiatric Genomics Consortium, 2013*). We confirmed that SHN-1's PDZ domain binds the EGL-19 carboxy-terminal motif (-VTTL$_{COOH}$) by both yeast two-hybrid and GST-pull down assays (*Figure 1—figure supplement 1A and B*). Thus, like their mammalian counterparts, SHN-1 binds to EGL-19/ Ca$_v$1 (*Zhang et al., 2005*).

### EGL-19/Ca$_v$1 calcium currents are diminished in *shn-1* mutants

The impact of Shank binding on Ca$_v$1.3 function has been assessed by over-expression in Xenopus oocytes and in cultured neurons (*Stanika et al., 2016*; *Zhang et al., 2005*) but has not been tested in mutant or transgenic animals. Because *shn-1* is expressed in muscles (*Stefanakis et al., 2015*), we assayed EGL-19/Ca$_v$1 channel function by recording voltage-activated calcium currents in body muscle, using salines containing potassium channel blockers (TEA and 4AP). In these conditions, the remaining voltage-activated inward current is entirely blocked by the EGL-19 antagonist nemadipine (*Lainé et al., 2011*). In *shn-1(tm488)* null mutants, calcium current density was significantly decreased (*Figure 1B,C,E*). Neither the voltage-dependence of calcium current activation (*Figure 1D*) nor the deactivation kinetics (*Figure 1F*) were altered in *shn-1(tm488)* null mutants. The *shn-1* calcium current defect was rescued by a transgene restoring SHN-1 expression in body muscles (*Figure 1E*), confirming that SHN-1 has a cell autonomous effect on EGL-19/Ca$_v$1 currrent. To determine if SHN-1's effects on calcium currents were specific, we measured voltage-activated potassium currents in body muscles (*Figure 1—figure supplement 1C*). Neither the voltage-dependence nor the current density of fast and slow potassium currents were significantly altered in *shn-1* mutants. Collectively,

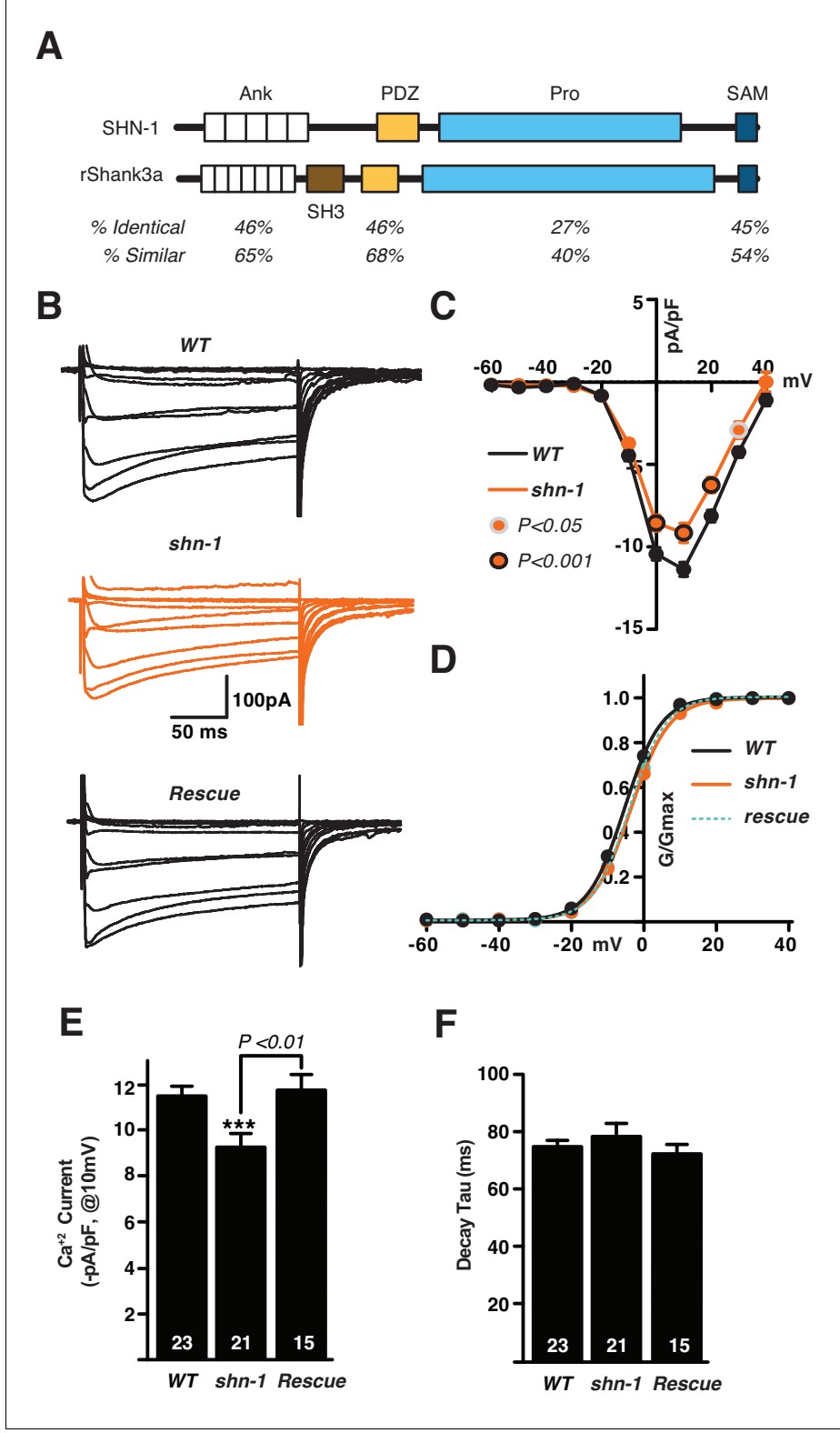

**Figure 1.** SHN-1 promotes EGL-19/Ca$_v$1 channel function. (A) The protein domains found in SHN-1 and rat Shank3A are compared. SHN-1 lacks an SH3 domain but contains all other domains found in mammalian Shank proteins. Homology between the worm and mammalian protein is shown for each domain. (B–F) Voltage-activated Ca$^{+2}$ currents were recorded from adult body wall muscles of the indicated genotypes at holding potentials of −60 to +40 mV. Averaged traces (B), mean current density as a function of holding potential (C), normalized

*Figure 1 continued on next page*

*Figure 1 continued*

conductance as a function of holding potential (D), mean current density at 10 mV (E), and mean deactivation time constants (F) are shown. *shn-1* mutants had significantly decreased $Ca^{+2}$ current-density and this defect was rescued by a single copy transgene expressing SHN-1 in body muscles (*nuSi26*) (D). No significant differences were observed for voltage-dependence of current activation and de-activation kinetics. The number of animals analyzed is indicated for each genotype. Values that differ significantly from wild type controls are indicated (***$p<0.001$). Error bars indicate SEM. Mean, standard errors, sample sizes, and p values for this figure are shown in **Supplementary file 1**.

The following figure supplement is available for figure 1:

**Figure supplement 1.** Supplemental data related to *Figure 1*.

these results suggest that SHN-1 specifically regulates the expression or function of EGL-19/$Ca_v$1 channels.

## SHN-1 binding to EGL-19 increases $Ca_v$1 current density

SHN-1 effects on calcium current density could result from direct binding of SHN-1 to EGL-19/$Ca_v$1 or indirectly via other SHN-1 binding partners. We did several experiments to distinguish between these possibilities. First, we utilized CRISPR to isolate two deletion alleles that alter the EGL-19 carboxy-terminus (*nu495* and *nu496*) (*Figure 2A*). Both deletion mutants exhibited decreased calcium current density (similar to the defect observed in *shn-1* null mutants) (*Figure 2B–C*). The *egl-19 (nu496)* mutation had no effect on the voltage-dependence of calcium current activation nor on deactivation kinetics (*Figure 2—figure supplement 1A and B*). The *egl-19(nu496)* and *shn-1* null mutations did not have additive effects on calcium current density in double mutants, as would be predicted if SHN-1's effects on calcium current require direct binding to EGL-19's carboxy-terminus (*Figure 2D and E*). To further examine the functional impact of the SHN-1 PDZ interaction with EGL-19, we analyzed *shn-1(ok1241)* mutants, which have an in-frame deletion spanning exons encoding the PDZ domain and part of the proline-rich domain (*Figure 1—figure supplement 1D*). The *shn-1 (ok1241)* mutants exhibited a decrease in calcium current density similar to those observed in *shn-1* null and the *egl-19* carboxy-terminal deletion mutants and had no effect on voltage-dependence of current activation nor on deactivation kinetics (*Figure 2—figure supplement 1C–F*). Collectively, these results suggest that SHN-1 binding to EGL-19's carboxy-terminus promotes the expression or function of L-type calcium channels.

## SHN-1 promotes trafficking of EGL-19 channels to the cell surface

SHN-1 effects on calcium current could result from a change in EGL-19 delivery to the cell surface. We performed two further experiments to test this idea. First, we measured the gating currents of voltage-activated channels in body muscles (*Figure 3A–B*). The activation of $Ca_v$ channels is mediated by depolarization-induced movements of positively charged residues in membrane-spanning S4 helices, which are termed gating charges. When there is no net calcium current (by holding the muscle membrane at the reversal potential), gating charge movement can be measured as a small voltage-activated current. The magnitude of gating currents can be used as a measure of $Ca_v$ channel surface abundance (*Fu et al., 2011*; *Hulme et al., 2006*). The total voltage activated gating charge in body muscles was significantly reduced in *shn-1* null mutants (*Figure 3B*). This *shn-1* mutant defect in gating charge is unlikely to result from decreased surface delivery of other voltage-activated channels because voltage-activated potassium currents were unaltered in *shn-1* mutants (*Figure 1—figure supplement 1C*). Thus, analysis of gating currents suggests that the decreased calcium current exhibited in *shn-1* mutants arises from decreased trafficking of EGL-19/$Ca_v$1 channels to the cell surface.

To further investigate SHN-1's effects on EGL-19 trafficking, we designed a fluorescent reporter construct containing a trans-membrane domain fused to EGL-19's cytoplasmic tail domain (504 amino acids), which includes the carboxy-terminal PDZ ligand (*Figure 3C*). This chimeric protein (designated Terrier to indicate the presence of EGL-19's cytoplasmic tail) was expressed in body muscles. The Terrier protein contains tagRFP in the cytoplasmic domain, and pHluorin (a pH-

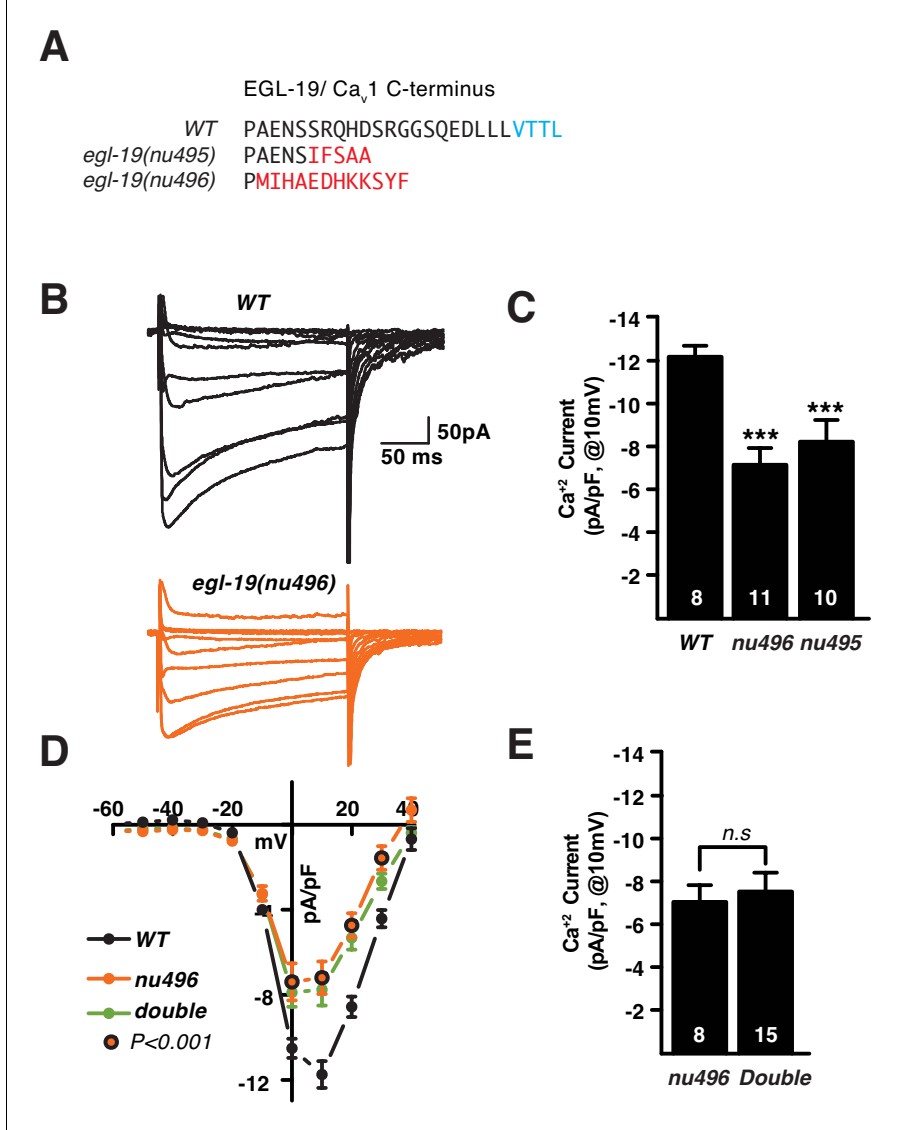

**Figure 2.** SHN-1 binding to EGL-19's carboxy-terminus promotes the expression or function of L-type calcium channels. (**A**) Predicted c-terminal sequences of mutant EGL-19 proteins are shown. *egl-19(nu496)* is a 22 bp deletion and *egl-19(nu495)* is a 5 bp deletion, both resulting in frame shifts that delete the carboxy-terminal PDZ ligand of EGL-19 (-VTTL$_{COOH}$). Residues in blue represent the PDZ ligand. Residues in red represent those introduced by the frame shift mutations. (**B–E**) Voltage-activated Ca$^{+2}$ currents were recorded from adult body wall muscles of the indicated genotypes at holding potentials of −60 to +40 mV. Representative traces (**B**), mean current density at 0 mV (**C, E**), and mean current density as a function of holding potential (**D**) are shown. The *egl-19(nu496)* and *shn-1(tm488)* single mutants had similar decreases in Ca$^{+2}$ current-density, and additive defects were not observed in the double mutant. The number of animals analyzed is indicated for each genotype. Values that differ significantly from wild type controls are indicated (***p<0.001). Error bars indicate SEM. Mean, standard errors, sample sizes, and p values for this figure are shown in **Supplementary file 1**.

The following figure supplement is available for figure 2:

**Figure supplement 1.** Supplemental data related to Figure 2.

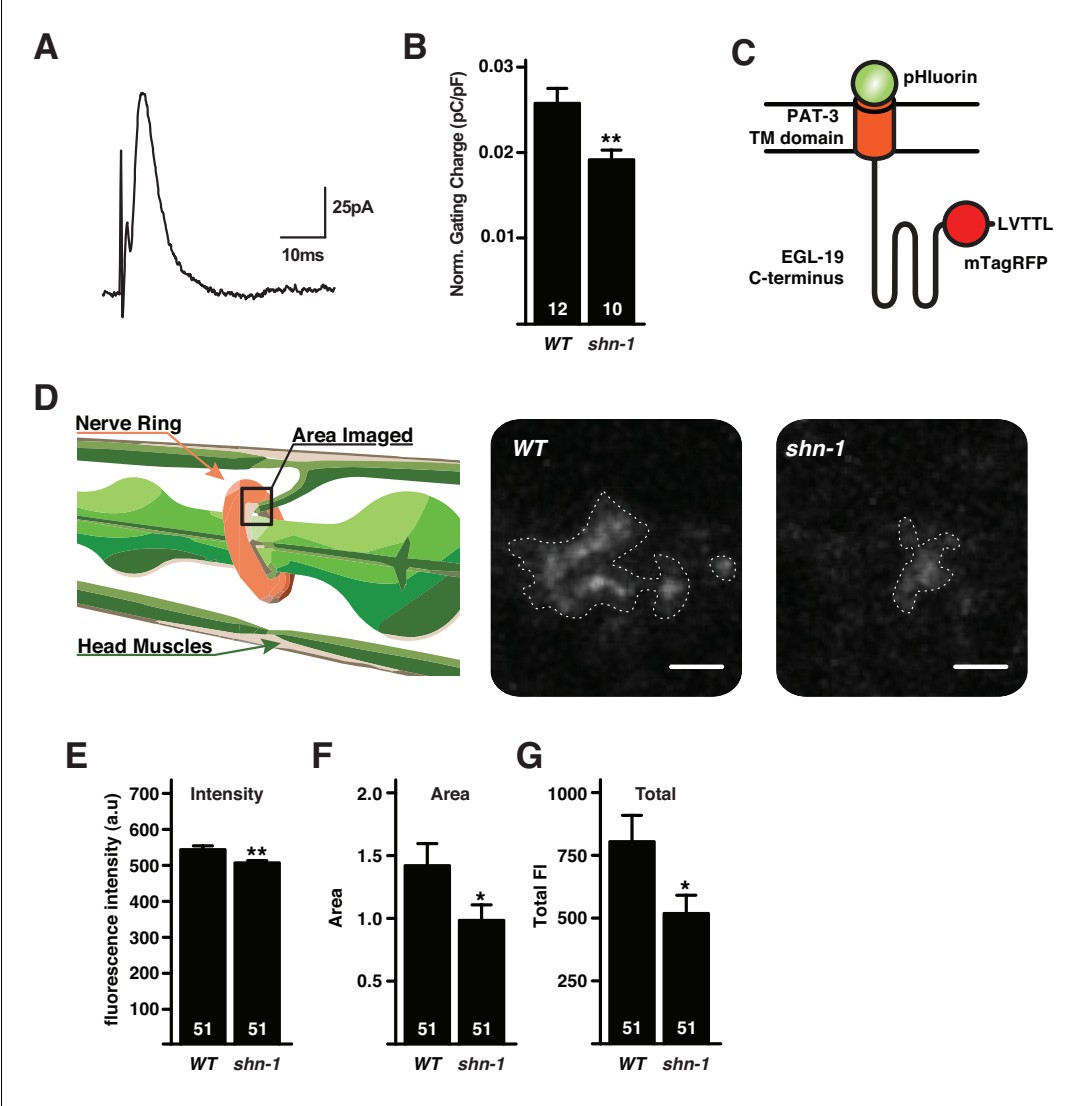

**Figure 3.** SHN-1 promotes EGL-19/Ca$_v$1 delivery to the cell surface. (A–B) Voltage-activating gating currents were significantly decreased in *shn-1* null mutants. Averaged trace of gating current in wild type adult body muscles (A) and mean gating charge (normalized to capacitance) (B) are shown. (C–G) Surface delivery of the Terrier fusion protein is significantly reduced in *shn-1* null mutants. (C) A schematic illustrating the structure of the Terrier fusion protein is shown. (D) A schematic illustrating the imaged region (left) and representative images of Terrier pHluorin fluorescence in the nerve ring are shown. Mean pHluorin puncta intensity (E), pHluorin puncta area (F), and total pHluorin puncta fluorescence (G) are shown. Regions of interest utilized to quantify Terrier fluorescence are indicated (D). The number of animals analyzed is indicated for each genotype. Values that differ significantly from wild type controls are indicated (**p<0.01, *p<0.05). Error bars indicate SEM.

sensitive GFP) in the ectodomain (*Figure 3C*). Thus, red fluorescence reports total Terrier protein while green fluorescence reports surface Terrier molecules. When expressed in body muscles, Terrier exhibits a punctate green and red fluorescence in the nerve ring, where body muscles receive synaptic input (*Figure 3C–D*). In *shn-1* null mutants, the intensity and size of green Terrier puncta were significantly decreased (*Figure 3D–G*), indicating decreased surface Terrier protein in the nerve ring. Thus, decreased calcium current density (20%) in *shn-1* mutants was mirrored by similar decreases in gating charge (26%) and total surface Terrier fluorescence (36%). Collectively, these results suggest that the decreased calcium current in *shn-1* null mutants arises from decreased delivery of EGL-19/Ca$_v$1 channels to the cell surface.

## *gem-4* is an activity-induced CRH-1/CREB target expressed in muscles

Increased cytoplasmic calcium activates expression of a large number genes, hereafter designated activity-induced gene expression. Although $Ca_v1$ channels account for a small fraction of bulk calcium entry in neurons, $Ca_v1$ channels account for the majority of activity-induced gene expression (*Ma et al., 2012*). This privileged ability of $Ca_v1$ channels to activate gene expression is thought to be mediated by direct physical coupling of $Ca_v1$ channels to the calcium sensors responsible for activating CREB (*Deisseroth et al., 1996*; *Wheeler et al., 2008*).

Because *shn-1* mutations alter EGL-19/$Ca_v1$ current, we hypothesized that SHN-1 may also play a role in activity-induced gene expression. To test this idea, we first identified activity-induced muscle genes. We analyzed gene expression following depolarization of body muscles with a nicotinic acetylcholine (ACh) agonist (levamisole, Lev). Lev-induced genes were identified using the Affymetrix *C. elegans* gene chip. This analysis identified 427 genes whose expression was significantly increased following muscle depolarization (>2 fold change, FDR p<0.05) (*Figure 4*, *Supplementary file 2*). 67% (287/427) of Lev-induced genes contain binding sites for the myogenic transcription factor HLH-1 (<5 kb from the transcriptional start site, TSS) in chromatin-immunoprecipitation experiments (http://www.modencode.org), suggesting that these genes are expressed in body muscles. Of the HLH-1 binding genes, 81% (233/287) contain predicted CREB binding sites (<5 kb from the TSS). These results suggest that *C. elegans* body muscles (like other excitable cells) have a large number of activity-induced genes, many of which are potential CREB transcriptional targets.

Using this Lev-induced gene list, we devised a simple reporter assay for CREB induced gene expression. For this purpose, we focused on the *gem-4*/Copine gene, which was the top hit from our analysis of Lev-induced genes (induced ~30 fold) (*Figure 4*). The *gem-4* promoter contains

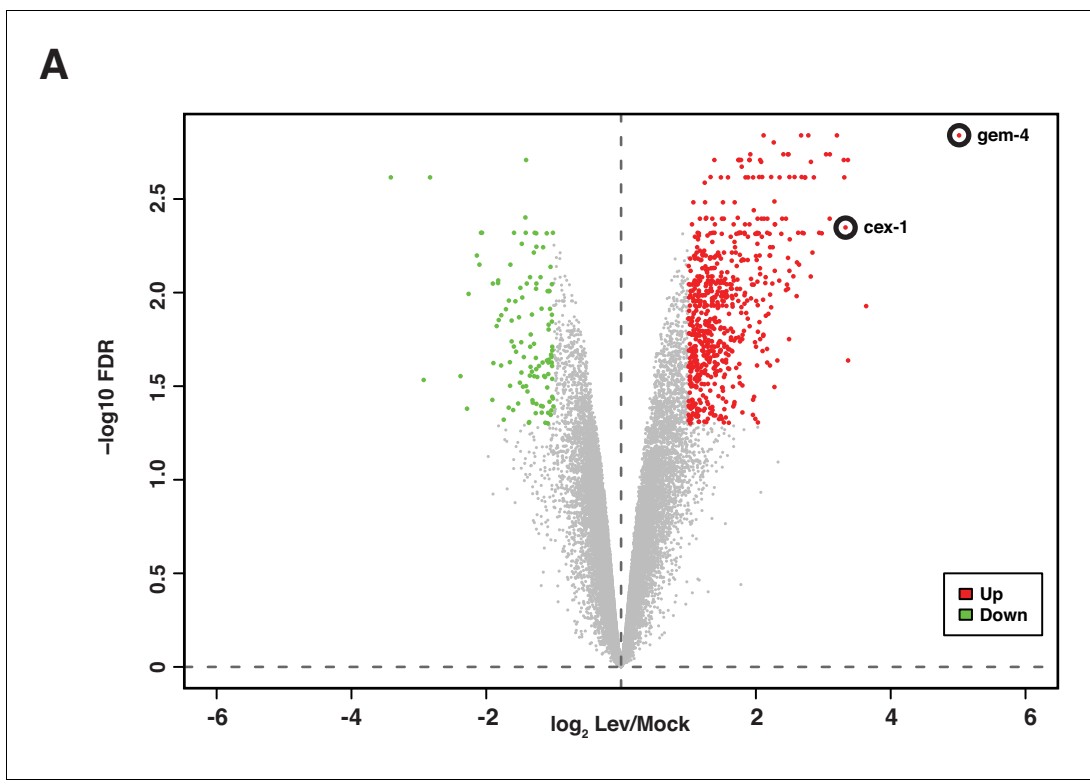

**Figure 4.** Analysis of mRNA abundance following muscle depolarization. mRNA abundance in Lev (200 µM, 1 hr) versus mock treated synchronized L4 larvae is plotted. Fold change (x-axis) is plotted against the statistical significance (y-axis) for each probeset. Fold changes are shown in $\log_2$ scale. Adjusted P values are shown in -$\log_{10}$ scale. Genes with increased (red dots) and decreased (green dots) expression are indicated (>2 Fold-change, FDR p<0.05). Probe sets corresponding to *gem-4* and *cex-1* are indicated. All genes that are differentially expressed following Lev treatment are listed in *Supplementary file 2*.

multiple CRH-1/CREB binding sites, implying that it could be a direct CRH-1 transcriptional target. Quantitative RT-PCR confirmed that Lev treatment increased *gem-4* mRNA levels 8-fold (*Figure 5A*). We designed a transcriptional reporter (*Figure 5B*) that compares expression of the *gem-4* promoter (expressing NLS-GFP) with a control promoter unaffected by depolarization (the *myo-3* promoter, expressing NLS-mCherry) in individual muscle cells. Using this reporter, we found that Lev treatment increased *gem-4* expression 8–12-fold (*Figure 5C–D*) while *myo-3* expression was unaltered (*Figure 5—figure supplement 1A*). Lev-induction of the *gem-4* reporter was eliminated by mutations inactivating a Lev receptor subunit (UNC-29), indicating that *gem-4* induction was not mediated by an off target effect of Lev (*Figure 5C*). Lev-induction of *gem-4* was also blocked in mutants lacking CRH-1/CREB and this defect was rescued by a transgene expressing CRH-1 in body muscles (*Figure 5D*). These results identify *gem-4* as a CRH-1/CREB target expressed in body muscles.

Lev treatment is a non-physiological stimulus that produces prolonged muscle depolarization. To determine if synaptic activity induces the *gem-4* reporter, we evoked excitatory synaptic input to muscles by photo-stimulating transgenic animals expressing channel rhodopsin in cholinergic motor neurons (*Figure 5E*). In patch clamp recordings of body muscles, spontaneous action potentials are observed at ~1 Hz (*Gao and Zhen, 2011*). Therefore, to mimic a realistic pattern of synaptic input, we photo-stimulated motor neurons (25 ms light pulse) at 1–20 Hz for 20 min. The *gem-4* reporter was significantly induced following 2, 5, 10, and 20 Hz photo-stimulation (*Figure 5E*). Induction was not observed at lower frequencies (0.5 and 1 Hz) nor when animals were cultured without all trans-retinal (ATR) (*Figure 5E*). Photo-induction of *gem-4* expression was eliminated in *unc-13* mutants (*Figure 5—figure supplement 1B*), which have dramatically reduced synaptic vesicle exocytosis (*Richmond et al., 1999*). By contrast, *unc-13* mutations did not prevent Lev-induced *gem-4* expression (*Figure 5—figure supplement 1C*). Similar levels of *gem-4* induction were produced by 20 Hz photo-stimulation and Lev treatment (*Figure 5—figure supplement 1B–C*). Thus, the *gem-4* reporter was induced by both synaptic and Lev-evoked muscle depolarization. Expression of a mouse *gem-4* paralog (N-copine) in hippocampal neurons is also induced by high frequency stimulation of acute brain slices (*Nakayama et al., 1998*); consequently, activity-induced copine expression is observed in both muscles and neurons and is conserved across phylogeny.

## SHN-1 is required for *gem-4* induction

Using this *gem-4* reporter construct, we next asked if SHN-1 is required for CRH-1/CREB-induced gene expression. Lev-induced *gem-4* expression was significantly decreased in *shn-1(tm488)* null mutants (*Figure 5F*). To determine if SHN-1 controls expression of other activity-induced genes, we developed a transcriptional reporter for a second Lev-induced gene (*cex-1*) (*Figures 4* and *5I*). Expression of the *cex-1* reporter in body muscles was dramatically induced following Lev treatment (*Figure 5J*). Because baseline *cex-1* expression in untreated muscles could not be reliably detected, we were unable to accurately measure the fold-induction of the *cex-1* reporter following Lev treatment. As seen with the *gem-4* reporter, we found that Lev-induced *cex-1* expression in muscles was dramatically reduced in *shn-1* null mutants (*Figure 5J*). Taken together, these results support the idea that SHN-1 promotes activity-induced gene expression in body muscles.

Next we asked if *gem-4* induction requires binding of SHN-1's PDZ domain to EGL-19's carboxy-terminus. Deleting EGL-19's PDZ ligand [in *egl-19(nu496)* mutants] significantly increased *gem-4* induction while deleting SHN-1's PDZ domain [in *shn-1(ok1241)* mutants] had no effect on *gem-4* induction (*Figure 5G–H*). These results indicate that SHN-1 regulates EGL-19/Ca$_v$1 current and CRH-1/CREB activation by distinct mechanisms, since the former requires PDZ binding to EGL-19 while the latter does not.

## Calcium current and *gem-4* induction are sensitive to *shn-1* gene dose

Deletion and duplication of human shank genes are both associated with ASD, schizophrenia, and mania (*Bonaglia et al., 2006*; *Durand et al., 2007*; *Failla et al., 2007*; *Gauthier et al., 2010*; *Han et al., 2013*). If Shank CNVs are causally associated with these psychiatric disorders, cellular phenotypes should be similarly sensitive to Shank copy number. To test this idea, we analyzed the effect of *shn-1* gene dosage on calcium currents and Lev-induced *gem-4* expression (*Figure 6*). We analyzed animals with 0 (*tm488* homozygotes), 1 (*tm488/+* heterozygotes), 2 (WT), and 4 (WT +2

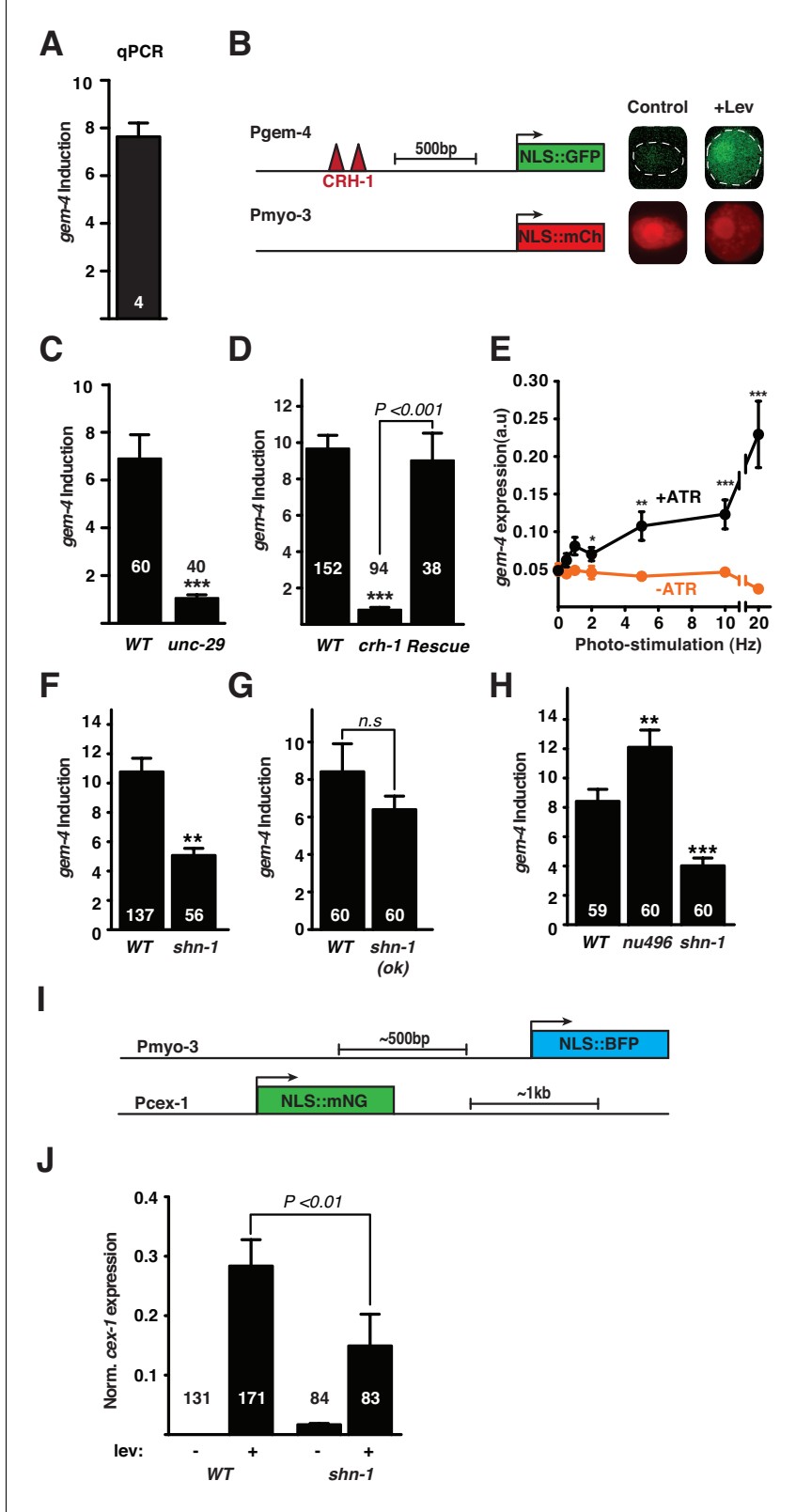

**Figure 5.** *gem-4* Copine expression in body muscle is induced by depolarization. Induction of *gem-4* expression was analyzed by qPCR (**A**) and using a transcriptional reporter (**B–H**). (**A**) The abundance of *gem-4* mRNA (assessed by qPCR) was increased following 1 hr levamisole (Lev) exposure. The number of biological replicates is indicated. (**B**) A schematic diagram of the *gem-4* reporter construct (left) and representative images of muscle

*Figure 5 continued on next page*

*Figure 5 continued*

nuclei (right) before and after a 20 min Lev exposure, and 2 hr recovery. (C–D) The mean fold induction of the *gem-4* reporter (P*gem-4*) after Lev treatment is shown. Lev-induced *gem-4* expression was abolished in mutants lacking UNC-29, an essential subunit of the Lev receptor (C) and in mutants lacking the transcription factor CRH-1 (D). The *crh-1* mutant defect in *gem-4* induction was rescued by a transgene expressing CRH-1 in body muscles (D). (E) Expression of the *gem-4* reporter was measured following photo-stimulation of transgenic animals that express ChR2 in cholinergic motorneurons. Expression of *gem-4* was significantly increased by 2, 5, 10, and 20 Hz photo-stimulation (for 20 min). Photo-evoked *gem-4* expression was not observed when animals were not cultured with ATR. (F–G) The fold induction of the *gem-4* reporter following Lev exposure was significantly reduced in *shn-1* null mutants (F) but not in *shn-1(ok1241)* mutants, which lack the PDZ domain (G). Lev-induced *gem-4* expression was significantly increased in *egl-19(nu496)* mutants, which lack the carboxy-terminal PDZ ligand (H). (I–J) Lev induction of the *cex-1* reporter is significantly reduced in *shn-1* mutants. (I) Schematics of the *cex-1* and *myo-3* reporters are shown. (J) Expression of the *cex-1* reporter (normalized to *myo-3* expression in the same nucleus) was significantly increased by Lev treatment. The Lev-induced expression of the *cex-1* reporter was significantly reduced in *shn-1* mutants. The number of animals analyzed is indicated for each genotype. Values that differ significantly from wild type controls are indicated (***$p<0.001$; **$p<0.01$; *$p<0.05$). Error bars indicate SEM.
The following figure supplement is available for figure 5:

**Figure supplement 1.** Supplemental data related to *Figure 5*.

single copy *shn-1* transgenes) copies of *shn-1*. Compared to wild type controls, muscle calcium current density was significantly increased in animals containing 1 and 4 copies of *shn-1* and was significantly decreased in animals containing 0 copies of *shn-1* (*Figure 6A–C*). The kinetics of calcium current deactivation were not significantly altered by changes in *shn-1* dosage (*Figure 6D*). Similarly, when compared to wild type controls, *gem-4* induction was significantly diminished in animals containing 0, 1, and 4 *shn-1* copies (*Figure 6E*). Thus, L-type calcium current density and *gem-4* induction were both sensitive to *shn-1* copy number. Interestingly, increased and decreased *shn-1* gene dosage produced similar defects in calcium current and *gem-4* induction.

## Cholinergic transmission is not sensitive to *shn-1* gene dosage

To determine SHN-1's role in synaptic transmission, we recorded excitatory post-synaptic currents (EPSCs) from body muscles. Evoked responses were significantly larger in *shn-1* null mutants (*Figure 7A,C*). The shape of evoked responses in wild type and *shn-1* mutants were indistinguishable, indicating that the kinetics of evoked release was unaltered (*Figure 7B*). The rate and amplitude of spontaneous miniature EPSCs (mEPSCs) were unaltered in *shn-1* mutants (*Figure 7—figure supplement 1A–C*). The change in evoked EPSC amplitude combined with unaltered mEPSC amplitudes indicates a pre-synaptic change in cholinergic transmission. The *shn-1* null EPSC defect was rescued by transgenes expressing SHN-1 in body muscles, implying that SHN-1 functions post-synaptically (*Figure 7B*). The evoked EPSCs (*Figure 7C*) and mEPSCs (*Figure 7—figure supplement 1D–E*) observed in animals with 1, 2, and 4 *shn-1* copies were not significantly different, indicating that synaptic transmission is not sensitive to *shn-1* copy number. Similar results were recently reported in mice where glutamatergic transmission in the striatum was enhanced in $Shank3B^{-/-}$ homozygotes but this effect was not observed in $Shank3B^{+/-}$ heterozygotes (*Peixoto et al., 2016*).

## Discussion

Because Shank proteins are highly enriched at post-synaptic densities, prior studies focused on the idea that Shank regulates synapse formation or function in some manner (*Jiang and Ehlers, 2013*). Here we provide evidence that Shank regulates Ca$_v$1 calcium current density and activity-induced gene expression, and that these two cellular functions are mediated by distinct dosage-sensitive mechanisms.

## SHN-1 promotes EGL-1/Ca$_v$1 channel delivery to the cell surface

We find that SHN-1's PDZ domain binds the EGL-19/Ca$_v$1 carboxy-terminus (like their mammalian counterparts) and that disrupting this interaction decreased muscle calcium current. Mutations

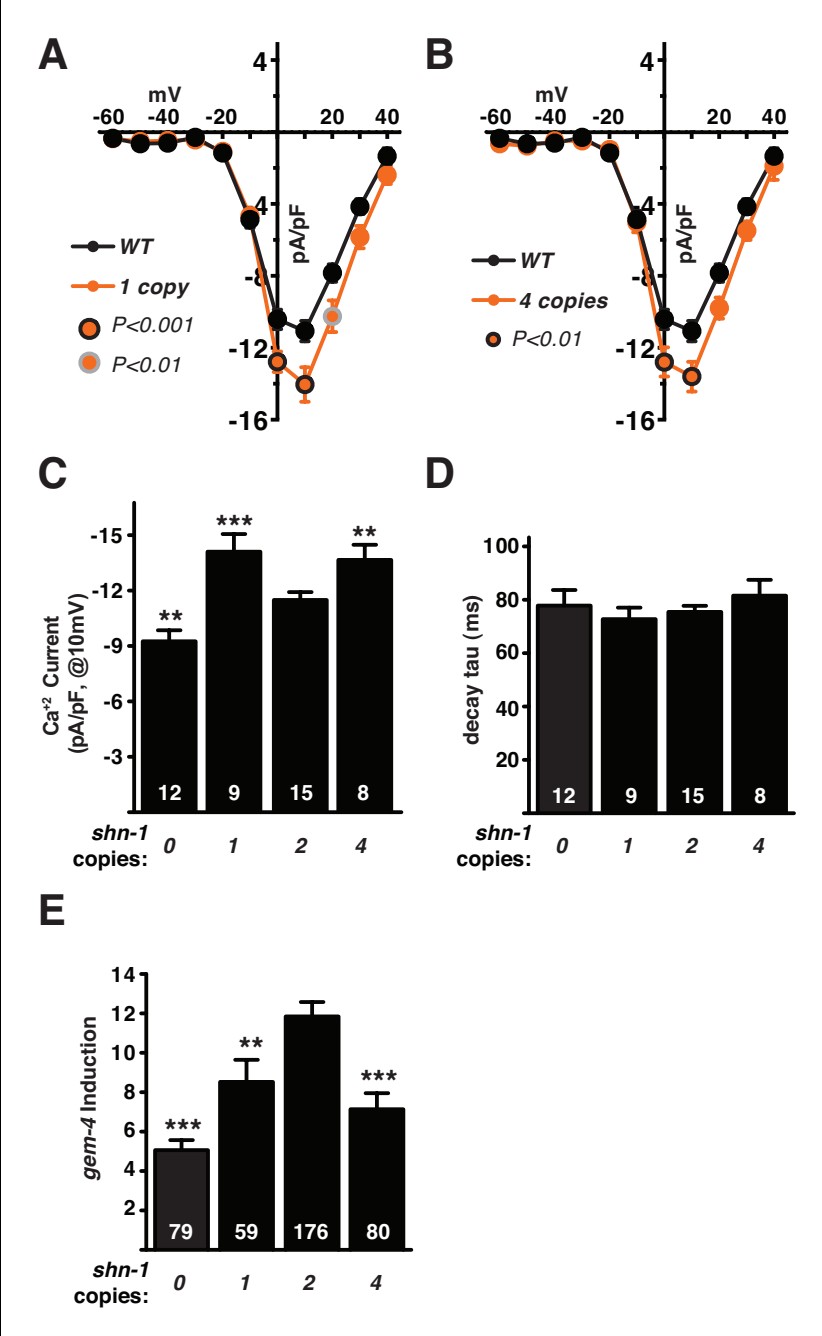

**Figure 6.** *shn-1* gene dosage regulates calcium current density and *gem-4* induction. The effect of varying *shn-1* gene dosage on calcium current density (**A–B**) and *gem-4* induction (**C**) was analyzed. The following genotypes were analyzed: 0 copies of *shn-1* [*shn-1(tm488)* homozygotes], 1 copy of *shn-1* [*shn-1(tm488)/+* heterozygotes], 2 copies of *shn-1* (wild-type) and 4 copies of *shn-1* (*nuSi26* homozygotes in wild-type). (**A–B**) Muscle Ca$^{+2}$ current was sensitive to changes in *shn-1* gene dose, with decreased (0 *shn-1* copies) and increased (1 and 4 *shn-1* copies) current density observed in the indicated genotypes. Mean current density as a function of holding potential (**A**), mean current density at 10 mV (**B**), and mean current deactivation time constants (**C**) are shown. (**D**) Lev-induced *gem-4* expression was significantly reduced in animals with 0, 1, and 4 copies of *shn-1*. The number of animals analyzed is indicated for each genotype. Values that differ significantly from wild type controls are indicated (***p<0.001; **p<0.01). Error bars indicate SEM. Mean, standard errors, sample sizes, and p values for panels A-C are shown in ***Supplementary file 1***.

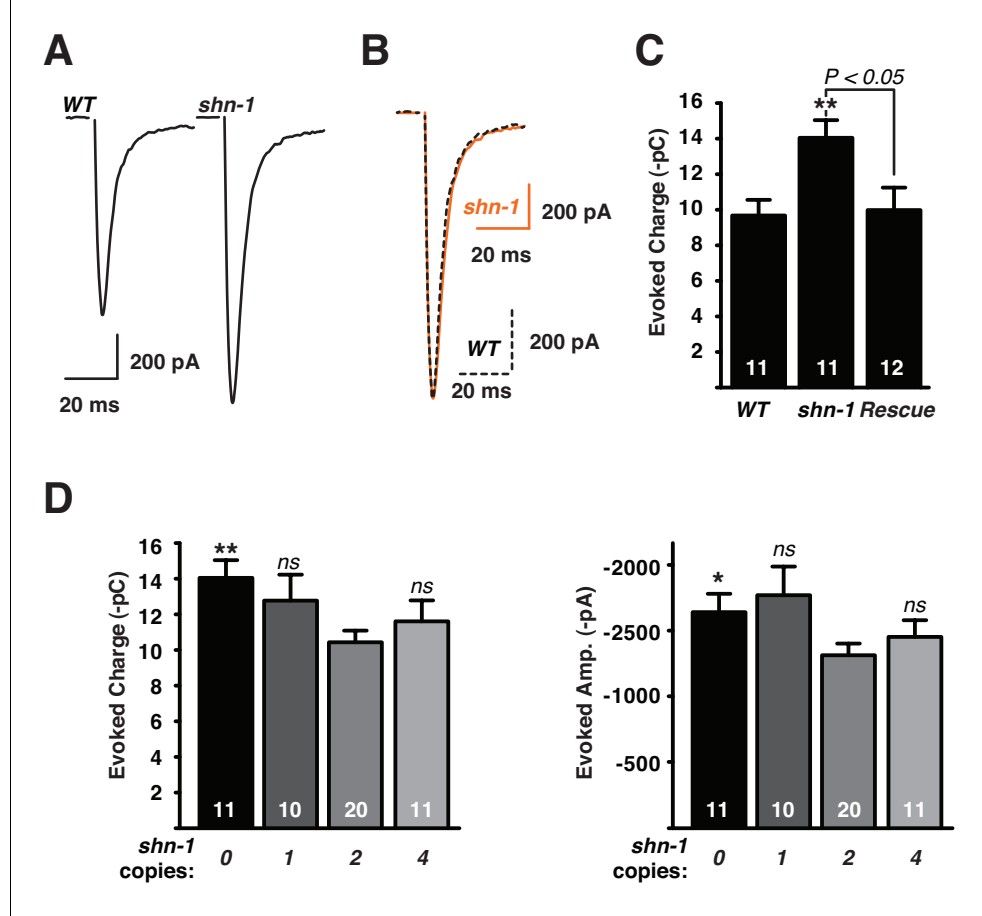

**Figure 7.** Synaptic transmission at the NMJ is not sensitive to *shn-1* gene dosage. Stimulus-evoked EPSCs were recorded from adult body wall muscles. Representative traces of evoked responses (**A**), averaged *shn-1* and WT evoked responses (normalized to equal amplitudes) (**B**), and mean evoked charge transfer (**C**) are shown. Evoked charge was significantly increased in *shn-1* homozygotes and this defect was rescued by a single copy transgene (*nuSi26*) that restored SHN-1 expression in body muscles (**C**). Averaged peak normalized *shn-1* and WT evoked responses have indistinguishable rise and decay times, indicating that rise and decay kinetics were unaltered (**B**). Evoked charge transfer (**D**) and peak amplitudes (**E**) did not differ significantly in animals containing 1, 2, and 4 *shn-1* copies. The number of animals analyzed is indicated for each genotype. Values that differ significantly from wild type controls are indicated (**p<0.01; ns, not significant). Error bars indicate SEM. Mean, standard errors, sample sizes, and p values for this figure are shown in *Supplementary file 1*.
The following figure supplement is available for figure 7:

**Figure supplement 1.** *shn-1* mutations and gene dosage had no effect on mEPSCs.

deleting SHN-1's PDZ domain and those deleting EGL-19's PDZ ligand both decreased EGL-19 calcium current and additive defects were not observed in double mutants. These results suggest that SHN-1's effects on calcium current are mediated by binding of its PDZ domain to EGL-19's carboxy-terminus. Analysis of gating currents and Terrier reporter fluorescence suggest that the decreased muscle calcium currents in *shn-1* mutants arises primarily from decreased trafficking of EGL-19/$Ca_v$1 channels to the plasma membrane.

Shank effects on $Ca_v$1 current in mammalian neurons or muscles have not been reported. The carboxy-terminus of mammalian $Ca_v$1.3 (-ITTL$_{COOH}$) and $Ca_v$1.2 (-VSSL$_{COOH}$) channels are both predicted to bind PDZ domains (*Zhang et al., 2005*). When mutant constructs lacking the carboxy-terminal PDZ ligands of $Ca_v$1.3 or $Ca_v$1.2 channels were expressed in hippocampal neurons, no changes in calcium current were observed; however, the $Ca_v$1.3 truncation mutant exhibited

decreased synaptic localization (*Weick et al., 2003*; *Zhang et al., 2005*). The rat Shank 1 and 3 PDZ domains exhibit strong binding to $Ca_v1.3$ (but not $Ca_v1.2$). Thus, Shank proteins could promote synaptic targeting of $Ca_v1.3$; however, the PDZ protein responsible for $Ca_v1.3$ targeting has not been identified. These studies relied on over-expression of mutant $Ca_v1$ channels (rather than knockin mutations altering the endogenous genes); consequently, it remains possible that Shank proteins could also regulate $Ca_v1$ current in mammalian neurons or muscles.

## SHN-1 promotes activity-induced expression of *gem-4*

Prior studies of mammalian neurons proposed that $Ca_v1$ channel binding to a PDZ scaffolding protein promotes activity-induced CREB phosphorylation (*Weick et al., 2003*; *Zhang et al., 2005*). These studies showed that deleting the carboxy-terminal PDZ ligand in $Ca_v1.2$ or 1.3, or over-expressing peptides containing the PDZ-ligands diminish activity-induced CREB phosphorylation and transcription of CREB targets. The PDZ protein responsible for these effects was not identified in these studies. Here we show that SHN-1 promotes expression of a CRH-1/CREB transcriptional target in muscles but that this effect does not require binding of SHN-1's PDZ domain to EGL-19's carboxy-terminus. In fact, deleting EGL-19's PDZ ligand significantly enhanced depolarization induced CRH-1/CREB target expression, perhaps by disrupting EGL-19 interaction with another protein.

Our results suggest that SHN-1's effects on calcium current density and *gem-4* induction are mediated by distinct mechanisms. Increased calcium current is mediated by SHN-1's PDZ domain binding to EGL-19's carboxy-terminus whereas another SHN-1 domain (most likely the Ankyrin or Proline-rich domains) enhances CREB activation.

A puzzling aspect of our results is that calcium current density and induced *gem-4* expression were poorly correlated. For example, reduced *gem-4* induction was observed in mutants with decreased (*shn-1* null mutants) and increased (*shn-1/+* heterozygotes) EGL-19/$Ca_v1$ current density. Similar results were also reported in cultured mammalian neurons, where CREB phosphorylation was poorly correlated with calcium current density and instead was better correlated with $Ca_v1$ open probability (*Wheeler et al., 2008*). Based on these results (and others), these authors proposed that CREB phosphorylation is mediated by a calcium sensor that is spatially very close to (and possibly physically associated with) activated $Ca_v1$ channels (*Deisseroth et al., 1996*; *Wheeler et al., 2008*). In this scenario, CREB phosphorylation would be strongly correlated with calcium levels in the nanodomain of a $Ca_v1$ channel but less correlated with global cytoplasmic calcium. Thus, our results suggest that CRH-1 activation may also be mediated by tight coupling of a calcium sensor to activated EGL-19/$Ca_v1$ channels.

## Implications for understanding psychiatric disorders

In humans, Shank deletions and duplications both confer risk for ASD and schizophrenia (*Bonaglia et al., 2006*; *Durand et al., 2007*; *Failla et al., 2007*; *Gauthier et al., 2010*; *Han et al., 2013*). Thus, too little or too much Shank is associated with psychiatric traits. If Shank CNVs are causally linked to these psychiatric disorders, cellular defects linked to ASD should exhibit similar sensitivity to Shank gene dosage. Consistent with this idea, several studies previously reported cellular and behavioral defects in Shank3[m/+] heterozygotes (*Han et al., 2016*; *Orefice et al., 2016*; *Wang et al., 2016*; *Yi et al., 2016*). Increased Shank3 copy number has been analyzed in a single study (*Han et al., 2013*), which reported several behavioral and synaptic phenotypes. None of these studies identified phenotypes that were shared between decreased and increased Shank3 dosage. In most cases, it was not determined if cellular defects were cell autonomous consequences of altered Shank3 dosage (*Han et al., 2013*; *Orefice et al., 2016*; *Wang et al., 2016*; *Zhou et al., 2016*). Here we find that changes in *shn-1* gene dosage alter two cell autonomous phenotypes in muscles: L-channel calcium current and expression of CREB target genes. For both phenotypes, similar defects were observed in muscles with decreased and increased *shn-1* dosage. Thus, we identify two cellular phenotypes that exhibit the same pattern of dose sensitivity that is observed for Shank3 in schizophrenia and ASD. Based on these results, we propose that human Shank CNVs also cause cellular phenotypes, which may include altered calcium current and CREB target expression. By contrast, other *shn-1* mutant phenotypes (e.g. evoked EPSCs) were not sensitive to *shn-1* gene dosage. Thus, sensitivity to Shank copy number provides a useful criterion to determine which phenotypes are more likely to contribute to the psychiatric traits associated with Shank CNVs. Two recent studies

showed that Shank3 binds directly to $I_h$ and TRPV1 channels in neurons and that the corresponding currents densities are significantly reduced in Shank3$^{+/-}$ heterozygotes (*Han et al., 2016*; *Yi et al., 2016*). These results (together with those reported here) suggest that an important function of Shank proteins is to regulate ion channel densities.

How does *shn-1* dose alter L-current and CREB target expression? In general, dosage sensitive phenotypes are thought to occur by disrupting the function of multimeric protein complexes. In such cases, under and over-expression of individual subunits alter the stoichiometry of assembled holo-complexes, leading to decreased activity. For example, increased and decreased expression of the yeast histones H2A and H2B results in similar loss of function phenotypes (chromosome loss and altered gene expression) (*Clark-Adams et al., 1988*; *Meeks-Wagner and Hartwell, 1986*). Shank binds many other synaptic proteins (*Lee et al., 2011*; *Sakai et al., 2011*) and undergoes zinc induced polymerization into large complexes (*Baron et al., 2006*; *Hayashi et al., 2009*). Thus, increased and decreased Shank abundance could disrupt the stoichiometry of post-synaptic complexes. Our results suggest that L-current and CREB activation are sensitive to subtle changes in the stoichiometry of Shank protein complexes. Our results further suggest that dynamic changes in the composition of Shank complexes provides a mechanism to regulate circuit function and plasticity (and potentially psychiatric traits).

Shank regulation of L-currents and CREB activation could both contribute to the pathophysiology of ASD. Human Shank mutations are particularly linked to ASD associated with intellectual disability (*Leblond et al., 2014*). CREB has long been linked to learning and memory in several model organisms (*Bourtchuladze et al., 1994*; *Dash et al., 1990*; *Yin et al., 1994*); consequently, Shank's CREB activation function could directly contribute to cognitive deficits associated humans Shank CNVs. CREB-induced BDNF expression promotes development of inhibitory synapses in the cortex (*Hong et al., 2008*). Thus, Shank's CREB activation function could alter synaptic inhibition, a phenotype found in several ASD models (*Dani et al., 2005*; *Rubenstein and Merzenich, 2003*). Finally, L-channels play an important role in calcium signaling in dendrites and spines (*Higley and Sabatini, 2012*). Thus, Shank's Ca$_v$1 current density function could contribute to cognitive or developmental defects in ASD by adjusting dendritic calcium signaling in CNS neurons. For these reasons, we propose that Shank effects on Ca$_v$1 currents and CREB target expression could play an important role in the pathophysiology of ASD (and potentially other psychiatric disorders).

## Materials and methods

### Experimental procedures

#### Strains

Strain maintenance and genetic manipulation were performed as described (*Brenner, 1974*). Animals were cultivated at room temperature (~22°C) on agar nematode growth media seeded with OP50 bacteria. The following strains were used in this study:

KP7624 *nuIs525[Pgem-4::NLS-GFP; Pmyo-3::NLS-mCherry] V*
KP7598 *unc-29(x29) I; nuIs525 V*
KP7583 *crh-1(tz2) III; nuIs525 V*
KP7601 *oxSi91[Punc17::ChIEF::mCherry] II; nuIs525 V*
KP7618 *unc-13(s69) I; nuIs525 V*
KP7896 *unc-13(s69) I; oxSi91 II; nuIs525 V*
KP7032 *shn-1(tm488) II*
KP7272 *shn-1(ok1241) II*
KP7461 *nuSi26[Pmyo-3::shn-1] shn-1(tm488) II*
KP7573 *shn-1(tm488) II; nuIs525 V*
KP7574 *bli-2(e768) shn-1(tm488) II; nuIs525 V*
KP7567 *nuSi26 II; nuIs525 V*
KP7493 *nuSi26 II*
KP7212 *bli-2(e768) shn-1(tm488) II*
KP7992 *egl-19(nu496) IV*
KP7997 *egl-19(nu496) IV; nuIs525 V*
KP8046 *shn-1(tm488) II; egl-19(nu496) IV*

KP8047 *shn-1(tm488) II; egl-19(nu496) IV; nuIs525 V*
KP7991 *egl-19(nu495) IV*
KP8303 *nuSi74 [Pmyo-3::Terrier]*
KP8304 *nuSi74; shn-1(tm488)*
KP8274 *nuSi66 [Pmyo-3::NLS::TagBFP2]; nuSi67 [Pcex-1::NLS::mNeonGreen]; nuSi70[Pgem-4::NLS:: tagRFPt]*
KP8301 *nuSi66;nuSi67;nuSi70;shn-1(tm488)*

Transgenic animals were prepared by microinjection, and integrated transgenes were isolated following UV irradiation, as described (*Dittman and Kaplan, 2006*). Single copy transgenes were isolated by the MoSCI and miniMoS techniques (*Frøkjaer-Jensen et al., 2008*; *Frøkjær-Jensen et al., 2014*).

### *shn-1* dosage experiments

Animals with different *shn-1* copy numbers were constructed as follows: 0 copies, *shn-1(tm488)* homozygotes; 1 copy, WT males were crossed with *bli-2 shn-1(tm488)* homozygotes [KP7212 (for electrophysiology) and KP7574 (for *gem-4* induction)] and non-Blister F1 hermaphrodites were analyzed; 2 copies, WT males were crossed with *bli-2* homozygotes and non-Blister F1 hermaphrodites were analyzed; 4 copies, WT animals homozygous for the single copy transgene *nuSi26*. *gem-4* reporter expression in *bli-2/+* heterozygotes and WT controls did not differ significantly and were pooled for 2 copies of *shn-1*.

## Constructs and transgenes

### Pgem-4 reporter

2 kb of 5' non-coding sequences from the *gem-4* gene was cloned into a vector expressing NLS-GFP (pPD122.56). A *myo-3* promoter region (2.3 kb) was sub-cloned into a vector expressing NLS-mCherry. Both were injected into WT animals at 5 ng/µl and stable arrays picked. A single extrachromosomal array was integrated by UV irradiation and outcrossed six times (*nuIs525*).

### Pcex-1 reporter

KP#3310 has the *cex-1* promoter (1038 bp), a single SV40 NLS, mNeonGreen (*Shaner et al., 2013*) (codon optimized for *C. elegans*) followed by the EGL-13 NLS and the *cex-1* 3' UTR (2097bp) inserted between the SacII and PstI sites of pCFJ1662. A single hygromycin resistant integrant, *nuSi67* was obtained as described (*Frøkjær-Jensen et al., 2014*). *cex-1* reporter expression in each muscle nucleus was normalized to BFP expressed in the same nucleus (using the *myo-3* promoter).

### *myo-3* expression of BFP

KP#3309 has the *myo-3* promoter (2386 bp), a single SV40 NLS, mTagBFP2 (*Subach et al., 2011*) (codon optimized for *C. elegans*) followed by the EGL-13 NLS and the *unc-54* 3' UTR inserted between the SbfI and SnaBI sites of pCFJ901. A single G418 resistant integrant, *nuSi66* was obtained and mapped within the *abch-1* gene on Chromosome II as described (*Frøkjær-Jensen et al., 2014*).

### Terrier

KP#3308 has the *myo-3* promoter, PAT-4 signal sequence (from pPD122.36 Addgene), super-ecliptic pHluorin (*Dittman and Kaplan, 2006*), PAT-3 transmembrane domain (pPD122.36) followed by the cDNA of EGL-19B coding for residues 1374–1872, tagRFP-T (codon optimized for *C. elegans*), and the cDNA of EGL-19B coding for residues 1873–1877, followed by the *let-858* 3' UTR (pPD122.36) inserted between the HindIII and AflII sites in the polylinker of the miniMOS vector pCFJ910 (Addgene).

Single copy insertions of Terrier were obtained using the miniMOS technique as described (*Frøkjær-Jensen et al., 2014*).

### shn-1 rescue

A C33B4.3a cDNA was cloned using gateway into DONR221. Multisite gateway was used to assemble Pmyo-3::shn-1::unc54UTR into pCFJ150 and a single copy transgene (nuSi26) was obtained by injecting EG6699. The unc-119(ed3) allele was outcrossed from nuSi26 prior to analysis.

## Fluorescence imaging

Confocal imaging was performed using an Olympus 60x objective (NA 1.45) on an Olympus FV-1000 confocal microscope at 5x digital zoom. For Pgem-4 imaging, ~15 worms were exposed to 1 mM Lev for 20 mins. Two hours after Lev stimulation, worms were immobilized on 10% agarose pads with 0.3 µl of 0.1 µm diameter polystyrene microspheres (Polysciences 00876–15, 2.5% w/v suspension). Individual muscle nuclei were imaged next to the terminal bulb of the pharynx and analysed using FIJI (https://fiji.sc). The ratio was obtained of green (Pgem-4) to red (Pmyo-3) for each nucleus. A same day WT control (+/- Lev) was analyzed for all genotypes. For terrier imaging ~6–10 worms were immobilized on 10% agarose pads with 0.3 µl of 0.1 µm diameter polystyrene microspheres and for each worm the closest neuromuscular junction to the surface was imaged. Worms were only accepted for analysis if they had clearly identifiable red and green puncta. Both intensity and area were analysed using FIJI.

## Retinal plates

4 µl of all trans-retinal (ATR, 100 mM dissolved in ethanol) was mixed with 250 µl of OP50 E. coli and spread on 60 mm NGM plates. Plates were allowed to dry for 24 hr and approx. 40 L4 animals were added and allowed to grow in the dark for 16–24 hr before the assay. Control plates used 4 µl of ethanol mixed with 250 µl of OP50.

## Optogenetic stimulation

ACh release at NMJs was evoked in animals expressing the Channelrhodopsin variant ChIEF in cholinergic neurons (oxSi93) (*Watanabe et al., 2013*). Animals were photo-stimulated with seven 470 nm Rebel LEDs mounted on a 40 mm SinkPadII fitted with a Round Concentrator Lens and powered with a 700 mA DC Driver (Luxeon Star LEDs). 20 min of 25 ms light pulses were generated at the indicated frequencies using an Arduino Uno. Pulse frequency and duration were confirmed using a photodiode and oscilloscope.

## qPCR

Total RNA was purified from a synchronized population of young adult worms treated with 200 µM Levamisole for 1 hr and mock-treated samples. RNA was isolated using standard Trizol-bromochloropropane extraction methods in combination with the Qiagen RNeasy Kit. RNA was DNase treated using the Qiagen on-column RNase free DNase set. Samples were prepared from the following genotypes: wild-type (N2 Bristol) and shn-1 (tm488) on two separate days. One micrograms of total RNA was used to synthesize cDNA using RETROscript (Ambion). Real-time PCR was performed using iTaq Universal SYBR Green Supermix (BioRad) and a 7500 Fast Real-Time PCR System (Applied Biosystems). All reactions were run in triplicate and on at least two biological replicates. All the values are normalized to rpl-32 as internal control as well as to the transcript levels in untreated samples. Statistical significance was determined using the two-tailed Student's t test.

## Microarray analysis

RNA isolation and cDNA synthesis was performed as described for qPCR. 6 Affymetrix C. elegans GeneChip were used (3 × 1 hr 200 µM Levamisole exposure and 3x mock-treated samples). Expression values were determined using the Robust Multi-chip Average (RMA) method. Probe sets that showed a > 2 fold change between mock and levamisole treated, with an unadjusted p-value of <0.0001 were considered to be Levamisole responsive. HLH-1 chip-seq data was taken from modencode (http://www.modencode.org). The PWM used for analysis of crh-1 binding sites was from Homer (http://homer.ucsd.edu/homer/).

## Electrophysiology

Whole-cell patch-clamp measurements were performed using a Axopatch 200B amplifier with pClamp 10 software (Molecular Devices). The data were sampled at 10 kHz and filtered at 5 kHz. All recordings were performed at room temperature (~19–21°C)

### Evoked EPSCs

Worms were superfused in an extracellular solution containing (in mM) 127 NaCl, 5 KCl, 26 $NaHCO_3$, 1.25 $NaH_2PO_4$, 20 glucose, 1 $CaCl_2$ and 4 $MgCl_2$, bubbled with 5% $CO_2$, 95% $O_2$ at 20°C. Whole cell recordings were carried out at –60 mV using an internal solution containing 105 mM $CH_3O_3SCs$, 10 mM CsCl, 15 mM CsF, 4 mM $MgCl_2$, 5 mM EGTA, 0.25 mM $CaCl_2$, 10 mM HEPES and 4 mM $Na_2ATP$, adjusted to pH 7.2 using CsOH. Under these conditions, we only observed endogenous acetylcholine EPSCs. For endogenous GABA IPSC recordings the holding potential was 0 mV. All recording conditions were as described (*McEwen et al., 2006*). Stimulus-evoked EPSCs were stimulated by placing a borosilicate pipette (5–10 µm) near the ventral nerve cord (one muscle distance from the recording pipette) and applying a 0.4 ms, 30 µA square pulse using a stimulus current generator (WPI).

### $Ca^{+2}$ current *recordings*

The pipette solution contained (in mM): 140 CsCl; 10 TEA-Cl; 5 MgCl2; 5 EGTA; 10 Hepes, pH 7.2, with ~320 mosM CsOH. The extracellular solution contained (in mM): 140 TEA-Cl; 5 CaCl2; 1 MgCl2; 3 4-aminopyridine; 10 glucose; five sucrose; 15 Hepes, pH 7.4, with ~330 mosM CsOH. The voltage-clamp protocol consisted of –60 mV for 50 ms, –90 mV for 50 ms, test voltage (from –60 mV to +4 mV) 200 ms. Access resistance was continuously monitored, and ranged between 7 and 15 MΩ. Series resistance was not compensated. The voltage dependence of the $Ca^{+2}$ current density were fitted with the equation: $I(V) = Gmax(V – Vrev) / (\{1 + exp[(V0.5 – V) / k]\})$, where $I(V)$ is the density of the current measured, $V$ is the test pulse, $Gmax$ is the maximum conductance, $Vrev$ is the apparent reversal potential, $V0.5$ is the half-activation voltage, and $k$ is a steepness factor. The decay tau was well fit by a single exponential function and calculated from a test potential of 0 mV fitting the curve from the peak of the current till the end of the pulse.

### Gating currents

The pipet and bath solutions were as described for the calcium current recordings. To resolve gating currents leak and capacitive transients were subtracted using a p/4 protocol, and measured by applying a series of test pulses at 5s intervals from the holding potential of –90 mV to potentials between +40 mV and +50 mV in 2 mV increments and integrating the gating charge movement at the reversal potential for the ionic current.

### $K^+$ current recordings

The bath solution contained (in mM): NaCl 140, KCl 5, CaCl2 5, MgCl2 5, dextrose 11 and HEPES 5 (pH 7.2, 320 mOsm); and the pipette solution contained (in mM): KCl 120, KOH 20, Tris 5, CaCl2 0.25, MgCl2 4, sucrose 36, EGTA five and Na2ATP 4 (pH 7.2, 323 mOsm). The voltage-clamp protocol consisted of –60 mV for 50 ms, –90 mV for 50 ms, test voltage (from –60 mV to +60 mV) 1000 ms. $IK_{fast}$ was defined as the peak current after the capacitance transients, and $IK_{slow}$ was defined as the average current of the last 100 ms of each voltage step.

## Statistics

Data was assessed for a normal distribution using the D'Agostino-Pearson normality test and equality of variances using the F-Test. For comparisons of normally distributed data with equal variances a two-tailed Student's t-test was used. For all other comparisons the Mann–Whitney U test was used. For analysis of calcium current-voltage relationships, two-way ANOVA with Sidak's correction for multiple comparisons was utilized. All statistics were performed in GraphPad Prism 6 and significant differences are indicated as follows: *$p < 0.05$, **$p < 0.01$, and ***$p < 0.001$.

## Acknowledgements

We thank the following for strains, advice, reagents, and comments on the manuscript: *C. elegans* stock center, S Mitani, and members of the Kaplan lab. This work was supported by a postdoctoral fellowship from the Nancy Lurie Marks Family Foundation (EP), and by research grants to JK from the NIH (NS32196) and from the Simons Foundation for Autism Research (SF273555). Additional data described in the manuscript are presented in the supporting online material.

## Additional information

### Funding

| Funder | Grant reference number | Author |
|---|---|---|
| National Institute of Neurological Disorders and Stroke | NS32196 | Joshua M Kaplan |
| Simons Foundation | SF273555 | Joshua M Kaplan |
| Nancy Lurie Marks Family Foundation | | Edward Pym |

The funders had no role in study design, data collection and interpretation, or the decision to submit the work for publication.

### Author contributions

EP, Conceptualization, Investigation, Writing—original draft, Writing—review and editing; NS, DJS, QH, Investigation, Methodology; KLT-P, Investigation; AA, RS, Formal analysis, microarray data analysis; SN, Resources, Investigation; JMK, Conceptualization, Supervision, Writing—original draft, Writing—review and editing

### Author ORCIDs

Katherine L Thompson-Peer, http://orcid.org/0000-0002-4200-3870
Joshua M Kaplan, http://orcid.org/0000-0001-7418-7179

## Additional files

### Supplementary files

• Supplementary file 1. This table lists means, errors, and p values for all electrophysiology figures.

• Supplementary file 2. This table lists Affymetrix probe sets that are differentially expressed following levamisole treatment (>2 fold change, FDR p<0.05).

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
