## [Decision Letter]

[Editors’ note: this article was originally rejected after discussions between the reviewers, but the authors were invited to resubmit after an appeal against the decision.]

Thank you for submitting your work entitled "Shank is a dose-dependent regulator of Cav1 calcium current and CREB target expression" for consideration by *eLife*. Your article has been reviewed by three peer reviewers, one of whom is a member of our Board of Reviewing Editors, and the evaluation has been overseen by Richard Aldrich as the Senior Editor. The reviewers have opted to remain anonymous.

Our decision has been reached after consultation between the reviewers. Based on these discussions and the individual reviews below, we regret to inform you that your work will not be considered further for publication in *eLife*.

Summary of reviewer comments and criticism:

The manuscript by Pym and colleagues presents phenotypic and molecular/genetic insight into the function of Shank with potential relevance to the pathophysiology of disease. In particular, these insights include potential regulation of calcium currents, downstream transcription and Shank gene dosage effects, a property of human disease biology that has yet to be well-connected to any particular cellular phenotype. For all of these reasons, the reviewers share enthusiasm for the study. However, there are several areas of concern. First, there are technical improvements that should be made and which are clearly specified in the reviews. Second, all three reviewers felt that additional insight into the regulation of the calcium channels was warranted. The extent to which this is possible remains unclear, but should be addressed at some level. Third, reviewers expressed concern by lack of mechanism. After further discussion this focused on several issues. On the one hand, the reviewers expressed concern regarding the broader cellular context for the phenotypic observations. Specifically, to what extent are the observed changes in *gem-4* specific to this gene, versus part of much broader regulatory disruptions? Likewise, to what extent are other channels or ionic currents altered? This is something that may be known to the authors based on their electrophysiological and gene profiling experiments that have been performed to date. If not, these are straightforward additions that could be pursued. On the other hand, there were concerns regarding the connection of Shank gene dosage to the cellular phenotypes. It may not be possible, in this system, to take this analysis to the level of changes in protein abundance, in vivo. But additional information to connect gene dosage to phenotypic severity would go a long way to establishing a model. Finally, the paper reads as a series of phenotypic observations and lacks coherence. It is not obvious to the reviewers how to address this issue. But, this is clearly something that could clearly benefit from additional experimentation as well as a more lengthy treatment in the text.

*Reviewer #1:*

This paper begins with the prior observation that the *C. elegans* orthologue of Shank is expressed in muscle, enabling the authors to examine the effects of this disease associated protein on the synaptic transmission properties and gene expression profile of the *C. elegans* neuromuscular junction (NMJ). The NMJ is a powerful system in the worm, allowing a combination of powerful genetics (dose-dependence of shank) with analysis of ionic currents, in vivo. The authors provide evidence (stated largely in their own words) that "SHN-1's effects on calcium current are mediated by binding of its PDZ domain to EGL-19's carboxy-terminus." Their results suggest, "that SHN-1's effects on calcium current density and *gem-4* induction are mediated by distinct mechanisms. Increased calcium current is mediated by SHN-1's PDZ domain binding to EGL-19's carboxy-terminus whereas another SHN-1 domain (most likely the Ankyrin or Proline-rich domains) enhances CREB activation." They find that, "changes in shn-1 gene dosage alter two cellular phenotypes: L-channel calcium current and expression of CREB target genes. Thus, their "results support the idea that human Shank CNVs cause cellular phenotypes that include altered calcium current and CREB target expression." When taken together, these data amount to an interesting advance with relevance to the cellular underpinnings of disorders caused by dose-dependent effects on expression of Shank that could reasonably be related to the cause or expression of disease related neurological symptoms in human. The manuscript suffers a bit from the brevity of style, presumable a consequence of prior submission at a different journal. Inclusion of additional information, already in the authors possession, or readily attained, could add an important dimension to this work that would increase the study's impact (see below).

Major issues:

1) The authors identify *gem-4* in an expression-array analysis of Lev-treated NMJ. Since the analysis has been completed, it seems reasonable to request that the entire transcriptional profiling data set be deposited in *eLife* as part of this manuscript. Is *gem-4* one of several hundred up-regulated genes, or one of only a few? The implications are important for understanding the degree to which the identified signaling is a simplification of a much broader effect. For example, a major conclusion of this paper is that Shank regulates both Cav1 function and, simultaneously, gene expression. But, the way the paper is written, the effects on gene expression are limited to a single gene.

2) Are the mechanisms identified by the authors part of a larger ion channel disruption process, or might these data point to differential effects in different cell types? The authors cite recent work from the Sudhof lab linking Shank to Ih. The authors make an effective argument that their observed effects on calcium channels are conserved in other species, and potentially relevant to disease. I suspect that the authors have analyzed additional currents and my have an indication, yes or no, that other channels are also dis-regulated. It may not be necessary to define which channels are dis-regulated, but it would be important to know whether there is a change in outward currents, A-type or delayed rectifiers. Conversely, if there is no change in these additional macroscopic currents, it would underscore the specificity of the effects being observed.

3) A notable limitation of this system is the inability to associate changes in calcium currents with the synaptic localization and/or function of the calcium channel. It is not immediately clear if this limitation can be solved. However, the authors do suggest that the calcium micro-domain may be relevant in their Discussion section, making this a relevant issue even at these small muscle cells.

*Reviewer #2:*

The manuscript "Shank is a dose-dependent regulator of Cav1 Calcium current and CREB target expression" by Pym et al. using *C. elegans* as a model system attempts to uncover how Shank interactions and copy number variations impact psychiatric disorders. Based on the data and analysis presented here, the authors make the following conclusions

1) The *C. elegans* homolog of Shank, SHN-1 interacts with the *C. elegans* homolog of Cav1.3 ELG19 though a direct PDZ interaction to regulate Cav1.3 current density

2) SHN-1 PDZ interaction with ELG19 is not required for CREB based actvation of *gem-4*.

3) One copy of SHN-1 or 4 copies of SHN-1 lead to identical changes in SHN-1 effects on ELG19 currents and *gem-4* induction.

4) Loss of SHN-1 leads to increases in cholinergic transmission compared to wild-type animals, while one copy of SHN-1 or 4 copies of SHN-1 have no effect on cholinergic transmission.

While this manuscript is interesting in its current form and the findings that a single copy of SHN-1 phenocopies 4 copies of SHN- 1/Shank and that the ELG19/Cav1.3 PDZ interaction with SHN-1/Shank interaction regulates ELG19/Cav1.3 density are very novel there are major concerns that need to be addressed before the paper is suitable for publication in *eLife*.

Major concerns:

1) The authors demonstrate that SHN-1 regulates Calcium current density and claim that this is through the PDZ interaction. However, the authors have no insight to whether this is due to decreases in Cav1.3 levels or due open probability. Therefore the authors should provide some mechanistic insight how SHN-1 regulates ELG19 current density through this PDZ interaction. This is critical as it has been previously shown by Zhang et al., 2005 that the PDZ ligand in the mammalian Cav1.3 is dispensable for interacting with mammalian Shank.

2) There is no consistency between Figure 1 and Figure 2 in regards to wt Calcium current activation and densities. The maximum current in wild-type in Figure 1 is at 0 mV in Figure 2 is +10 mV while in Figure 2 while in Figure 1 and Figure 2 the maximum Calcium currents for the mutant is 0 mV. So based on the data presented in the manuscript, it remains unknown if loss of SHN-1 or the EGL19 mutants impact Calcium current activation. In addition, in Figure 2 average Calcium density is two-fold higher than in Figure 1. Why the disparity? Furthermore, the holding voltages are different between the two. This can lead the reader to believe that hyperpolarizing the terminal to -90 mV relieves some sort of inhibition. The authors should redo Figure 2 with the same holding potential as Figure 1.

3) The authors look at copy number of SHN-1 but never give insight into the mechanisms of copy number regulation and it remains largely a phenomenological finding. Getting a mechanistic insight in how SHN-1 copy number leads to various changes that are observed in the manuscript is critical. In particular, how does one copy of SHN-1 or four copies of SHN-1 lead to the identical phenotype in Calcium current density? At minimum, the authors should correlate SHN-1 protein expression levels to copy number. What happens if you put in 3 copies, or 8 copies of SHN-1?

4) The authors demonstrate that loss of SHN-1 leads to an increase in cholinergic transmission. This is interesting since this there is no change in mEPSC amplitude or frequency. Some insight into how SHN-1 loss effects only evoked release should be done. Does the loss of SHN-1 also impact GABAergic transmission in the same manner?

5) The mutant ok1241 is not an acceptable mutant to just look at PDZ function. This mutant deletes much of the Proline domain in SHN-1.

6) Based on the IV recordings, it appears the SHN-1 copy number and EGL19 PDZ mutants impact EGL19 inactivation. The authors should go back it look at the Calcium current deactivation kinetics and see if there is a difference.

*Reviewer #3:*

In this manuscript, Pym and colleagues have used *C. elegans* to study the Autism linked gene Shank-3. In their work, they decided to focus on the interaction between the Shank-3 ortholog, Shn-1, and the sodium channel subunit Egl-19, as its mammalian ortholog was shown to play a role in a rare form of ASD. Pym and colleagues first show that Egl-19 and Shn-1 can indeed bind, just like their mammalian orthologs. They find that diminished currents in muscles in Shn-1 mutants are likely mediated by defects in Egl-19 expression, localization or function. Then, the authors focus on another Shn-1 mediated process, which is activity regulated expression. They perform a screen to identify the genes upregulated upon neuronal activation of muscle and decided to focus on *gem-4*. They confirm that *gem-4* activation requires Shn-1 activity but not its binding to Egl-19 thereby suggesting both processes are mediated by independent mechanisms. Finally, they determine that both Egl19 dependent currents as well as *gem-4* expression are sensitive to Shn-1 gene dose, something that is know from human patients. While the findings are of interest, especially due to the importance of Shank-3 in autism, I am not sure this paper represents a major step forward in our understanding. The Egl-19 and *gem-4* parts don't fit together and the story remains unfocused. Importantly, the mechanisms by which Shn-1 affects either Egl-19 dependent current or *gem-4* transcription remain unanswered. Finally, how all this is or is not relevant to autism is unclear.

Specific comments:

1) Yeast two hybrid and GST pull down assays can, at best, indicate that two proteins CAN bind to each other (in these very artificial contexts) and definitely do not prove interaction in vivo. I would use more careful words. Probably beyond the scope of this paper but to show that they interact in vivo, even in the non-biochemical organisms such as *C. elegans*, one can perform experiments with tagged transgenic proteins and test their proximity in a wide range of strategies… Also, the experiments shown in Sup 1A+B are quite poorly controlled. So Egl19 does not bind the beads but does it bind other PDZ domains? In the well-referenced paper (Zhang et al., 2005) they performed many required controls to be able to suggest that the mammalian orthologs interact.

2) I was surprised that the authors did not attempt to describe the sub cellular localization of Egl-19 in WT and mutant worms… The best experiment would be to endogenously tag Egl-19 but even the second best – to follow the localization of transgenic Egl-19 was not performed. The PDZ binding motif has to remain on the C-terminal so one can N-terminally tag or tag C-terminally with the addition of the PDZ binding motif afterwards. This lies at the heart of how Shn-1 affects Egl-19 and is therefore a key question that remains unanswered in this work.

3) The activity dependent transcription experiment – I was surprised that the raw data are not shown. Is *gem-4* the top hit but one out of many or is it the only one? Ideally one would like to view this.

4) Figure 3—figure supplement 2 does not exist (or at least I could not find it).

Taken together, I think this work represents two interesting stories that are currently incomplete and perhaps unfortunately do not fit very well together.

[Editors’ note: what now follows is the decision letter after the authors submitted for further consideration.]

Thank you for resubmitting your work entitled "Shank is a dose-dependent regulator of Cav1 calcium current and CREB target expression" for further consideration at *eLife*. Your revised article has been evaluated by Richard Aldrich (Senior editor) and a Reviewing editor.

The manuscript has been improved but there are some remaining issues that need to be addressed before acceptance, as outlined below:

The study has been greatly strengthened by the addition of new data. The authors have included data acquired but not previously included in the manuscript. The authors have added new analyses of ion channel currents that may have co-varied and have improved the analysis of calcium channel electrophysiology as requested in the initial round of review. In addition, the authors have extended their study by providing new information regarding the mechanism of calcium channel modulation. Taken together, these advances address the major outstanding concerns raised in the initial round of review. There was considerable enthusiasm in the initial round of review for the ideas and approach.

A remaining issue is one of clarifying new Figure 3. The figure could be clarified to make it easier to understand the region of interest that is being visualized in panel D. Although there is a cartoon, the transition from the cartoon to the images in panel D could be improved.

---

## [Author Response]

[Editors’ note: the author responses to the first round of peer review follow.]

We are pleased that the reviewers and editors found our results interesting and significant. The review provides many helpful suggestions, which have greatly improved our paper. In response to these comments, we added several new experiments: 1) we repeated our analysis calcium current densities using a single protocol (responding to Review #2, comment 2); 2) we show that voltage-activated gating currents in body muscles are significantly reduced in *shn-1* null mutants (Figure 3), suggesting that SHN-1 regulates delivery of EGL-19/Cav1 to the cell surface; 3) we show that surface delivery of a chimeric membrane protein that contains EGL-19’s cytoplasmic tail (Figure 3) is also significantly reduced in *shn-1* null mutants (Figure 3); 4) we show that SHN-1 mutations do not alter other voltage-activated currents in body muscles (Figure 1—figure supplement 1), indicating that SHN-1 selectively regulates calcium current density; 5) we provide a full description of Levamisole-induced gene expression (Figure 4, Table S2); and 6) we show that SHN-1 is required for activity-induced expression of a second muscle gene (*cex- 1*), which supports the idea that SHN-1 controls expression of multiple activity-induced genes (Figure 5). These results, as well as complete responses to all reviewer comments, are detailed below.

*Reviewer #1:*

[…] Major issues:

*1) The authors identify gem-4 in an expression-array analysis of Lev-treated NMJ. Since the analysis has been completed, it seems reasonable to request that the entire transcriptional profiling data set be deposited in eLife as part of this manuscript. Is gem-4 one of several hundred up-regulated genes, or one of only a few? The implications are important for understanding the degree to which the identified signaling is a simplification of a much broader effect. For example, a major conclusion of this paper is that Shank regulates both Cav1 function and, simultaneously, gene expression.*

As requested, we now provide a complete description of the Lev-induced gene set:

Results section:

“We analyzed gene expression following depolarization of body muscles with a nicotinic acetylcholine (ACh) agonist (levamisole, Lev). […] These results suggest that *C. elegans* body muscles (like other excitable cells) have a large number of activity-induced genes, many of which are potential CREB transcriptional targets.”

*But, the way the paper is written, the effects on gene expression are limited to a single gene.*

Prompted by this comment, we now show that SHN-1 is also required for expression of a second activity-induced gene:

Results:

“To determine if SHN-1 controls expression of other activity-induced genes, we developed a transcriptional reporter for a second Lev-induced gene (*cex-1*) (Figure 4 and Figure 5). Expression of the *cex-1* reporter in body muscles was dramatically induced following Lev treatment (Figure 5). Because baseline *cex-1* expression in untreated muscles could not be reliably detected, we were unable to accurately measure the fold-induction of the *cex-1* reporter following Lev treatment. As seen with the *gem-4* reporter, we found that Lev-induced *cex-1* expression in muscles was dramatically reduced in *shn-1* null mutants (Figure 5).”

*2) Are the mechanisms identified by the authors part of a larger ion channel disruption process, or might these data point to differential effects in different cell types? The authors cite recent work from the Sudhof lab linking Shank to Ih. The authors make an effective argument that their observed effects on calcium channels are conserved in other species, and potentially relevant to disease. I suspect that the authors have analyzed additional currents and my have an indication, yes or no, that other channels are also dis-regulated. It may not be necessary to define which channels are dis-regulated, but it would be important to know whether there is a change in outward currents, A-type or delayed rectifiers. Conversely, if there is no change in these additional macroscopic currents, it would underscore the specificity of the effects being observed.*

Thanks for pointing this out. As requested we now provide analysis of voltage-activated outward currents, finding that they are not significantly altered in *shn-1* mutants. As the reviewer indicates, these results are important because they indicate that SHN-1 selectively alters muscle Ca_v_1 current and has no effect on voltage-activated outward currents:

“To determine if SHN-1’s effects on calcium currents were specific, we measured voltage- activated potassium currents in body muscles (Figure 1—figure supplement 1). Neither the voltage- dependence nor the current density of fast and slow potassium currents were significantly altered in *shn-1* mutants.”

*3) A notable limitation of this system is the inability to associate changes in calcium currents with the synaptic localization and/or function of the calcium channel. It is not immediately clear if this limitation can be solved. However, the authors do suggest that the calcium micro-domain may be relevant in their Discussion section, making this a relevant issue even at these small muscle cells.*

We agree that this was an important limitation of our original submission. In response to this comment (and related comments in Reviews 2 and 3), we provide two new experiments (analysis of muscle gating currents and trafficking analysis of a chimeric membrane protein). Both of these new experiments support the conclusion that SHN-1 promotes the delivery of EGL-19/Ca_v_1 channels to the plasma membrane.

These results greatly enhance the significance our findings and are described in the revised text as follows:

“SHN-1 effects on calcium current could result from a change in EGL-19 delivery to the cell surface. […] Collectively, these results suggest that the decreased calcium current in shn-1 null mutants arises from decreased delivery of EGL-19/Ca_v_1 channels to the cell surface.”

*Reviewer #2:*

*[…] Major concerns:*

*1) The authors demonstrate that SHN-1 regulates Calcium current density and claim that this is through the PDZ interaction. However, the authors have no insight to whether this is due to decreases in Cav1.3 levels or due open probability. Therefore the authors should provide some mechanistic insight how SHN-1 regulates ELG19 current density through this PDZ interaction. This is critical as it has been previously shown by Zhang et al., 2005 that the PDZ ligand in the mammalian Cav1.3 is dispensable for interacting with mammalian Shank.*

We agree that this was a significant limitation in our original submission. Prompted by this concern (and related comments in Reviews 1 and 3), we add two new experiments to address the mechanism for Shank alteration of L-current. First, we record voltage-activated gating currents in muscles, finding that the decreased calcium current in *shn-1* results from a decrease in the number of channels in the plasma membrane (Figure 3). Second, we find that shn-1 null mutants have decreased trafficking of a chimeric reporter protein to the plasma membrane, again suggesting that SHN-1 promotes delivery of EGL- 19/Ca_v_1 channels to the cell surface (Figure 3). These data are now described in the revised text, as follows:

“SHN-1 effects on calcium current could result from a change in EGL-19 delivery to the cell surface. […] Collectively, these results suggest that the decreased calcium current in shn-1 null mutants arises from decreased delivery of EGL-19/Ca_v_1 channels to the cell surface.”

*2) There is no consistency between Figure 1 and Figure 2 in regards to wt Calcium current activation and densities. The maximum current in wild-type in Figure 1 is at 0 mV in Figure 2 is +10 mV while in Figure 2 while in Figure 1 and Figure 2 the maximum Calcium currents for the mutant is 0 mV. So based on the data presented in the manuscript, it remains unknown if loss of SHN-1 or the EGL19 mutants impact Calcium current activation. In addition, in Figure 2 average Calcium density is two-fold higher than in Figure 1. Why the disparity? Furthermore, the holding voltages are different between the two. This can lead the reader to believe that hyperpolarizing the terminal to -90 mV relieves some sort of inhibition. The authors should redo Figure 2 with the same holding potential as Figure 1.*

Recordings in these two figures were obtained with different protocols. Prompted by this comment, we repeated all recording using a single protocol (which yields larger current densities in all genotypes). All of the original findings were replicated with these new recordings. We apologize for the confusion caused by our use of multiple protocols in the original submission.

*3) The authors look at copy number of SHN-1 but never give insight into the mechanisms of copy number regulation and it remains largely a phenomenological finding. Getting a mechanistic insight in how SHN-1 copy number leads to various changes that are observed in the manuscript is critical. In particular, how does one copy of SHN-1 or four copies of SHN-1 lead to the identical phenotype in Calcium current density? At minimum, the authors should correlate SHN-1 protein expression levels to copy number. What happens if you put in 3 copies, or 8 copies of SHN-1?*

We agree that it would be very exciting to provide a mechanism to explain how calcium current and CREB target expression are similarly disrupted by decreased and increased Shank gene dosage. Even without a mechanistic explanation for *shn-1* dosage effects, we believe that our paper represents a significant advance for the field. To our knowledge, these phenotypes are the only examples where the same cell autonomous defects have been linked to increased and decreased Shank copy number (which parallels the effects of Shank3 CNVs on risk for Autism and schizophrenia). Now that we have established specific cellular traits that exhibit sensitivity to *shn-1* copy number, it is possible to design experiments asking how this dose sensitivity arises. However, I think it is fair to say that this is a difficult problem that will require many additional experiments. In fact, I am not aware of any good examples where the detailed biochemical mechanism for a haplo-insufficient or duplication phenotype has been determined. (Please let me know if you know of good examples!) Thus, while I agree that this is an important future goal, I hope that the reviewers will agree that describing a mechanism to explain shn-1 dosage sensitivity could be addressed in future publications.

Prompted by this comment, we made several changes in our revised paper. First, we now clearly list examples where cellular defects were identified in Shank3^+/-^ heterozygotes and in transgenic animals containing an extra copy of Shank3:

Introduction:

“If Shank3 mutations and CNVs are causally associated with these psychiatric disorders, cellular and circuit phenotypes should also be sensitive to Shank3 copy number. […] While these studies identify cellular deficits associated with Shank3 CNVs, it remains unclear which Shank binding partners and cellular functions are responsible for psychiatric traits, nor why these traits are sensitive to both increased and decreased Shank gene dosage.”

Discussion:

“In humans, Shank deletions and duplications both confer risk for ASD and schizophrenia (Bonaglia et al., 2006; Durand et al., 2007; Failla et al., 2007; Gauthier et al., 2010; Han et al., 2013). […] In most cases, it was not determined if cellular defects were cell autonomous consequences of altered Shank3 dosage (Han et al., 2013; Orefice et al., 2016; Wang et al., 2016; Zhou et al., 2016).”

Second, we provide a model for how increased and decreased *shn-1* copy number could result in the same phenotypes in the revised Discussion:

Discussion:

“How does *shn-1* dose alter L-current and CREB target expression? In general, dosage sensitive phenotypes are thought to occur by disrupting the function of multimeric protein complexes. […] Our results further suggest that dynamic changes in the composition of Shank complexes provides a mechanism to regulate circuit function and plasticity (and potentially psychiatric traits).”

*4) The authors demonstrate that loss of SHN-1 leads to an increase in cholinergic transmission. This is interesting since this there is no change in mEPSC amplitude or frequency. Some insight into how SHN-1 loss effects only evoked release should be done.*

We agree that the increased evoked release in *shn-1* mutants is interesting and merits further study; however, this evoked release defect does not exhibit sensitivity to *shn-1* dosage. The NMJ transmission phenotype is included here to demonstrate that sensitivity to shn-1 dosage can be utilized to distinguish between different Shank phenotypes. Further experiments analyzing this recessive synaptic defect (although interesting) would not alter our conclusion that some phenotypes are dose sensitive while others are not. Consequently, we hope that the reviewers will agree that this phenotype can be further analyzed in a future study.

*Does the loss of SHN-1 also impact GABAergic transmission in the same manner?*

The mIPSC frequency was unaltered while mIPSC amplitudes were modestly decreased in *shn-1* null mutants. As detailed in our response to comment #4, we prefer to focus on phenotypes sensitive to shn-1 copy number in this paper. For this reason, we respectfully request that more detailed studies of NMJ defects in shn-1 mutants could be addressed in future studies.

*5) The mutant ok1241 is not an acceptable mutant to just look at PDZ function. This mutant deletes much of the Proline domain in SHN-1.*

We agree that an allele that selectively deletes the PDZ domain would be preferable and are attempting to isolate such an allele with Cas9 (no luck thus far). Although the *ok1241* allele deletes part of the proline rich domain, we do not believe that this undermines any of our conclusions. Voltage-clamp recordings of body muscles in *ok1241* exhibit a similar decrease in calcium current density to those observed in shn-1 null mutants and in *egl-19(nu496)* mutants (which deletes the c-terminal ligand for the SHN-1 PDZ domain). And shn-1(null) mutations and the egl-19(*nu496*) PDZ ligand mutation did not have additive effects on calcium current in double mutants, implying that SHN-1 alters calcium current via its effects on EGL-19’s c-terminus. Collectively, these results strongly support our conclusion that binding of SHN-1’s PDZ domain to EGL-19’s c-terminal PDZ ligand promotes expression of EGL-19 current. By contrast *gem-4* activation was not decreased after deleting EGL-19’s c-terminal PDZ ligand (i.e. *nu496*) nor after deleting SHN-1’s PDZ domain and part of its Pro-rich domain (i.e. *ok1241*). Thus, the CREB activation function of SHN-1 clearly does not require either the SHN-1 PDZ domain or its ligand (EGL-19’s c- terminus). Does the reviewer suggest that disrupting the proline-rich domain could somehow obscure an effect of the PDZ domain on *gem-4* induction?

We hope that the revised text explains these results and our interpretations more clearly:

“SHN-1 effects on calcium current density could result from direct binding of SHN-1 to EGL-19/Ca_v_1 or indirectly via other SHN-1 binding partners. […] Collectively, these results suggest that SHN-1 binding to EGL-19’s carboxy-terminus promotes the expression or function of L-type calcium channels.”

*6) Based on the IV recordings, it appears the SHN-1 copy number and EGL19 PDZ mutants impact EGL19 inactivation. The authors should go back it look at the Calcium current deactivation kinetics and see if there is a difference.*

As requested, we analyzed deactivation kinetics and found no difference, as shown in: Figure 1; Figure 2—figure supplement 1; and Figure 6.

*Reviewer #3:*

*[…] Finally, they determine that both Egl19 dependent currents as well as gem-4 expression are sensitive to Shn-1 gene dose, something that is know from human patients.*

I am not aware of any publication showing that Shank gene dosage alters either CaV1 current or CREB activated gene expression (in any organism including humans). If the reviewer is aware of such studies, please let us know. Identifying these as cell autonomous phenotypes sensitive to Shank gene dosage is the central and most significant finding of our study.

*While the findings are of interest, especially due to the importance of Shank-3 in autism, I am not sure this paper represents a major step forward in our understanding.*

We apologize if our original text did not clearly state the significance of our findings. Our principal claims for significance are as follows:

1) Shank gene copy number had not been previously linked to either CaV1 current or expression of CREB target genes (both of which are thought to be linked to Autism and other psychiatric disorders).

2) No prior studies identified cellular phenotypes that exhibit the unusual pattern of sensitivity to Shank copy number whereby too little and too much Shank produce similar phenotypes (which mirrors the effects of Shank copy number on risk for Autism and schizophrenia).

3) Most prior studies suggest that Shank’s principal role is to regulate synapse formation or function (whereas our data suggest that non-synaptic functions could be equally important).

4) Our study represents a significant advance in identifying the cellular consequences of increased Shank copy number, which was analyzed in only a single prior study (the transgenic Shank3 mouse in Han et al., 2013).

5) Altered CaV1 current and CREB activation are cell autonomous defects associated with Shank copy number changes. By contrast, cell autonomy was not demonstrated in several prior studies of dose sensitive Shank phenotypes (Han, Nature 2013; Wang, Nat. Comm. 2016; Zhou, Neuron 2016; Orefice, Cell 2016).

6) Shank is linked to CaV1 current and CREB activation by direct biochemical interactions (unlike many other Shank associated phenotypes).

7) Linking Shank copy number to changes in CREB activation could provide an explanation for why Shank3 mutations are disproportionately found in Autism associated with intellectual disability.

Prompted by this comment, we revised our paper to highlight the significance of our findings:

Discussion):

“Consistent with this idea, several studies previously reported cellular and behavioral defects in Shank3^m/+^ heterozygotes (Han et al., 2016; Orefice et al., 2016; Wang et al., 2016; Yi et al., 2016). […] Thus, we identify two cellular phenotypes that exhibit the same pattern of dose sensitivity that is observed for Shank3 in schizophrenia and ASD.”

*The Egl-19 and gem-4 parts don't fit together and the story remains unfocused.*

For several reasons, we argue that Shank effects on CaV1 current and CREB activated gene expression should be described in a single paper. First, the primary source of calcium activating CREB is mediated by activated CaV1 channels; consequently, it makes sense to analyze both phenotypes in a single paper. Second, changes in CaV1 channels and activity-induced gene expression have both been linked to Autism (and other psychiatric disorders), but neither has been directly linked to Shank copy number. Third, both phenotypes exhibit the same unusual sensitivity to Shank copy number that is observed for risk of Autism and Schizophrenia (i.e. too much and too little Shank cause the same defect). Fourth, no other phenotypes have been described that exhibit this same pattern of sensitivity to Shank copy number. And fifth, our analysis suggests that these two Shank functions can be dissociated by certain mutations, implying that they are mediated by distinct biochemical mechanisms. This contrast would not be as evident if the stories were separated.

Prompted by this comment, we revised our text to explain why CREB activation is linked to changes in L- channels function:

“Increased cytoplasmic calcium activates expression of a large number genes, hereafter designated activity-induced gene expression. Although Cav1 channels account for a small fraction of bulk calcium entry in neurons, Cav1 channels account for the majority of activity- induced gene expression (Ma et al., 2013). This privileged ability of Cav1 channels to activate gene expression is thought to be mediated by direct physical coupling of Cav1 channels to the calcium sensors responsible for activating CREB (Deisseroth et al., 1996; Wheeler et al., 2008).”

*Specific comments:*

*1) Yeast two hybrid and GST pull down assays can, at best, indicate that two proteins CAN bind to each other (in these very artificial contexts) and definitely do not prove interaction in vivo. I would use more careful words. Probably beyond the scope of this paper but to show that they interact in vivo, even in the non-biochemical organisms such as C. elegans, one can perform experiments with tagged transgenic proteins and test their proximity in a wide range of strategies… Also, the experiments shown in Sup 1A+B are quite poorly controlled. So Egl19 does not bind the beads but does it bind other PDZ domains? In the well-referenced paper (Zhang et al., 2005) they performed many required controls to be able to suggest that the mammalian orthologs interact.*

We agree that the Y2H and pull down assays show that a direct interaction is plausible and that direct binding studies would be a stronger argument. Binding (i.e. coIP) assays from worm extracts are possible (and we have performed such experiments in several prior papers). However, biochemical analysis of CaV1 channels is difficult in any prep. In fact, no prior study has attempted to comprehensively identify proteins associated with CaV1 channels. CaV1 channels are huge, poorly expressed proteins that are difficult to solubilize. We have attempted to detect tagged EGL-19 channels in worm extracts by IP’s and western blots with no success.

Given the difficulty associated with biochemical analysis of CaV1 channels, we instead relied on genetic results to support our conclusion that SHN-1 binds to and regulates EGL-19 channels in vivo. Our results are as follows: 1) very similar calcium current defects are observed in *shn-1(null)* mutants, *shn-1(ok1241)* mutants which lack the PDZ domain, and *egl-19(nu496)* mutants which lack the c-terminal ligand for SHN-1’s PDZ domain; 2) the calcium current defects observed in *shn-1(null)* and in *egl-19(nu496)* mutants are both cell autonomous (i.e. both proteins must be present in muscle cells to obtain normal calcium currents), consistent with these proteins altering calcium current by directly binding to each other; 3) *shn-1(null)* mutations and *egl-19(nu496)* mutations do not have additive effects on calcium current in double mutants, implying that SHN-1 alters calcium current in a manner that requires EGL-19’s c-terminal ligand for SHN-1’s PDZ domain; and 4) *shn-1* null mutations cause similar decreases in EGL- 19 calcium current and in the surface abundance of a chimeric membrane protein containing EGL-19’s cytoplasmic tail sequence (Terrier), indicating that EGL-19’s cytoplasmic tail domain is sufficient to confer dependence on SHN-1 for membrane trafficking. We hope that the reviewer will agree that the strength of these genetic results offsets the absence of more direct biochemical analysis of SHN-1 binding to EGL-19.

*2) I was surprised that the authors did not attempt to describe the sub cellular localization of Egl-19 in WT and mutant worms… The best experiment would be to endogenously tag Egl-19 but even the second best – to follow the localization of transgenic Egl-19 was not performed. The PDZ binding motif has to remain on the C-terminal so one can N-terminally tag or tag C-terminally with the addition of the PDZ binding motif afterwards. This lies at the heart of how Shn-1 affects Egl-19 and is therefore a key question that remains unanswered in this work.*

We agree that this was an important limitation of our original submission. In response to this comment (and related comments in Review 1), we provide two new experiments (analysis of muscle gating currents and trafficking analysis of a chimeric membrane protein). Both of these new experiments support the conclusion that SHN-1 promotes the delivery of EGL-19/Ca_v_1 channels to the plasma membrane. These results greatly enhance the significance our findings and are described in the revised text as follows:

“SHN-1 effects on calcium current could result from a change in EGL-19 delivery to the cell surface. We performed two further experiments to test this idea. […] Collectively, these results suggest that the decreased calcium current in shn-1 null mutants arises from decreased delivery of EGL-19/Cav1 channels to the cell surface.”

*3) The activity dependent transcription experiment – I was surprised that the raw data are not shown. Is gem-4 the top hit but one out of many or is it the only one? Ideally one would like to view this.*

As requested, we now provide a summary of the genome wide expression profile associated with Lev- induced muscle depolarization. The complete set of activity induced genes is listed in a supplemental excel file, and is summarized in the revised text as follows:

Results:

“We analyzed gene expression following depolarization of body muscles with a nicotinic acetylcholine (ACh) agonist (levamisole, Lev). […] These results suggest that *C. elegans* body muscles (like other excitable cells) have a large number of activity-induced genes, many of which are potential CREB transcriptional targets.”

*4) Figure 3—figure supplement 2 does not exist (or at least I could not find it).*

Thanks for pointing out this error. We deleted the citation of this figure in the text.

[Editors’ note: the author responses to the re-review follow.]

*[…] A remaining issue is one of clarifying new Figure 3. The figure could be clarified to make it easier to understand the region of interest that is being visualized in panel D. Although there is a cartoon, the transition from the cartoon to the images in panel D could be improved.*

As request by the Reviewing Editor, we have modified the schematic in Figure 3, to illustrate the location of the imaged region for our Terrier reporter construct.